# Scaling and Distilling Transformer Models for sEMG

**Nicholas Mehlman**[*]                                                    *nmehlman@usc.edu*
*Viterbi School of Engineering, University of Southern California*

**Jean-Christophe Gagnon-Audet**[*]                                        *jcaudet@meta.com*
*Meta FAIR*

**Michael Shvartsman**                                                     *mshvarts@meta.com*
*Meta FAIR*

**Kelvin Niu**                                                            *kniu@meta.com*
*Meta FAIR*

**Alexander H. Miller**[†]                                                 *ahm@meta.com*
*Meta FAIR*

**Shagun Sodhani**[†]                                                      *sodhani@meta.com*
*Meta FAIR*

**Reviewed on OpenReview:** *https://openreview.net/forum?id=hFPWThwUiZ*

## Abstract

Surface electromyography (sEMG) signals offer a promising avenue for developing innovative human-computer interfaces by providing insights into muscular activity. However, the limited volume of training data and computational constraints during deployment have restricted the investigation of scaling up the model size for solving sEMG tasks. In this paper, we demonstrate that vanilla transformer models can be effectively scaled up on sEMG data and yield improved cross-user performance up to 110M parameters, surpassing the model size regime investigated in other sEMG research (usually <10M parameters). We show that >100M-parameter models can be effectively distilled into models 50x smaller with minimal loss of performance (< 1.5% absolute). This results in efficient and expressive models suitable for complex real-time sEMG tasks in real-world environments.

## 1 Introduction

Recently, there has been growing interest in using surface electromyography (sEMG) in conjunction with powerful deep learning techniques to decode human muscle activity (e.g. Di Nardo et al., 2022; Gaso et al., 2021; Wimalasena et al., 2022; Buongiorno et al., 2021; Ozdemir et al., 2020). sEMG offers the potential for novel human-computer interfaces (HCIs), where user gestures or movements can serve as direct control input (CTRL-labs at Reality Labs, 2024). Advances in hardware (e.g. Lu et al., 2024; CTRL-labs at Reality Labs, 2024) have now made it feasible to reliably capture sEMG outside of a controlled clinical setting. Supported by these developments, deep learning methods have been applied to a variety of EMG tasks, including muscle activation detection (Di Nardo et al., 2022), (Wimalasena et al., 2022), gesture classification (Atzori et al., 2016), (He et al., 2018), (Zhang et al., 2023b), and speech recognition (Wand & Schmidhuber, 2016).

However, there are a few limitations in previous works. First, many introduce significant complexity such as bespoke deep learning architectures (Wanga et al., 2024; Zabihi et al., 2023; Putro et al., 2024; Chen

---

[*]equal contribution
[†]equal contribution

et al., 2023; Zhang et al., 2022; 2023a;b; Liu et al., 2024) or additional training objectives (Dai et al., 2023; Zeng et al., 2022). This can act as a barrier for usability by practitioners, as departing from simple and battle-tested recipes reduces both the robustness of available implementations and support available from the community. This is especially true for in-the-wild practitioners, typically neuroscientists, who may lack familiarity with deep learning systems. This limitation may be due to the restricted quantity and/or diversity of available training data (Li et al., 2021), as large-scale data is often viewed as a prerequisite for applying contemporary vanilla deep-learning methods to complex tasks. Lacking large EMG corpora, researchers may historically have relied on exploiting neuroscience domain knowledge (e.g., hand-designed features or auxiliary losses) to achieve good performance. A second consideration commonly unaddressed in previous works is the computational challenge associated with running an HCI in the wild. In particular, there are likely to be substantial constraints on the model size (small enough to run on an edge device) and inference time (fast enough for the system to be responsive). Finally, the ability to effectively generalize to unseen users is critical to successful deployment in real-world applications, yet evaluation on unseen users is often neglected in the existing literature.

In this paper, we take steps towards addressing these challenges:

1. We start by applying a simple convolution and transformer architecture (Schneider et al., 2019) on the emg2qwerty task (Sivakumar et al., 2024) and outperform previous CNN-based SOTA by 20% (absolute). Then, we show that the performance of the transformer can be further improved with model scale, enabling us to improve over the SOTA performance by an additional 5% (absolute) by increasing the number of parameters from 2.2M to 109M. This differs from other works, which mainly focus on smaller model scales <10M parameters(Wanga et al., 2024; Rahimian et al., 2021; Montazerin et al., 2023; Zabihi et al., 2023; Zhang et al., 2023a; Yang et al., 2024) with complex non-standard deep learning methodology (Wanga et al., 2024; Dai et al., 2023; Zabihi et al., 2023; Putro et al., 2024; Chen et al., 2023; Zhang et al., 2022; 2023a;b; Liu et al., 2024; Zeng et al., 2022).

2. We analyze simple logit-based distillation (Hinton et al., 2015) for various model compression factors ranging from no reduction to 180x parameter reduction. We show that training larger transformer models followed by distillation into smaller models substantially outperforms direct training of the small-sized transformer without distillation across all compression factors. Furthermore, we find that we can reduce the parameter count of the transformer model by up to 50x before observing significant performance degradation ($< 1.5\%$ absolute).

3. We focus on performance on *cross-user* generalization, unlike most previous works in sEMG processing which often focus on reporting performance on cross-session generalization for the 'seen' (during training) users. Additionally, these works often use the Ninapro databases as evaluation, which may have insufficient data quantity/quality for contermporary deep learning, and/or insufficient task complexity to serve as a representative measure of real-world performance (CTRL-labs at Reality Labs, 2024; Saponas et al., 2008; Yang et al., 2024).

Put together, these contributions provide a simple and pragmatic roadmap for practitioners to apply in real-world settings. First, vanilla transformers can be scaled up beyond the sizes explored in other works. This is crucial to maximize cross-user performance, which is a prerequisite for the wider adoption of EMG models. Second, the simplest form of distillation is effective for producing scaled models that fit into specific compute constraints with minimal performance degradation (up to 50x). Thus, we can train models that maximize the tradeoff between capacity and performance. We believe that these findings represent an important step towards practical, efficient, and deployable EMG models amenable to in-the-wild usage.

The code used for training and distilling the models is available at `https://github.com/facebookresearch/fairemg`. We hope that it will make it easier for the scientific community to reproduce our results and extend this work.

| Reference | Task | Number of participants | Model |
|---|---|---|---|
| (She et al., 2010) | Lower-limb movt. | 3 | SVM |
| (Alkan & Günay, 2012) | Upper-arm movt | Not Reported | SVM |
| (Atzori et al., 2016) | Hand gesture | 78 | CNN |
| (Wand & Schmidhuber, 2016) | Speech recog. | 4 dev, 7 eval | DNN + HMM |
| (He et al., 2018) | Hand gesture | 27 | LSTM + MLP |
| (Cai et al., 2018) | Facial expr. | 7 | SVM |
| (Xia et al., 2018) | 3D limb motion est. | 8 | CNN + RNN |
| (Shioji et al., 2018) | Auth., hand gesture | 8 | CNN |
| (Morikawa et al., 2018) | Authentication | 6 | CNN |
| (Ozdemir et al., 2020) | Hand gesture | 30 | CNN (ResNet-50) |
| (Rahimian et al., 2021) | Hand gesture | 40 | Transformer |
| (Gaso et al., 2021) | Myopathy, ALS det. | 25 | FC-DNN |
| (Godoy et al., 2022) | Hand gesture | 10 | VIT |
| (Di Nardo et al., 2022) | Muscle activation | 18 + 30 | FC-DNN |
| (Chen et al., 2023) | Finger joint est. | 12 | Transformer |
| (Zabihi et al., 2023) | Hand gesture | 40 | Transformer |
| (Zhang et al., 2023b) | Hand gesture | 20 | LSTM + Transf. |
| (Liu et al., 2024) | Hand gesture | 50 | CNN + VIT |
| (Rani et al., 2024) | Hand gesture | 8 + Not Reported + 6 | RF, KNN, LDA |
| (Putro et al., 2024) | Finger joint est. | 5 | Transformer |
| (Eddy et al., 2024) | Hand gesture | 612 | RNN |
| (Sivakumar et al., 2024) | Typing | 108 | TDS-ConvNet |

Table 1: Prior work: most datasets are relatively small, and often classical ML approaches or older neural network architectures are used. When the number of participants is of the format X + Y, different devices or protocols were used so that they cannot be trivially combined into one dataset. In some cases, the number of participants is not reported in the papers, which we denote as 'Not Reported'. Most of these prior works have not evaluated their models on unseen users.

## 2 Background

This section summarizes related works, including sEMG modeling research in Section 2.1 and deep learning knowledge distillation in Section 2.2. For a tabular breakdown of related techniques vs our own, see the supplement (Table 6).

### 2.1 Surface Electromyography

The human central nervous system initiates muscular activity by transmitting an electrical impulse along a nerve bundle (Plonsey & Barr, 2007; CTRL-labs at Reality Labs, 2024). Surface electromyography (sEMG) uses external electrodes to measure these electrical action potentials as they propagate from the nerve fiber to the motor unit (Mokhlesabadifarahani & Gunjan, 2015; CTRL-labs at Reality Labs, 2024). sEMG data is noisy and non-stationary (Chowdhury et al., 2013; Cochrane-Snyman et al., 2016), making it a difficult signal modality for machine learning tasks.

Due to the difficulty of collecting EMG data, most open-source EMG datasets are small in terms of users and total recording time. Furthermore, most focus on capturing isolated movements that are relatively distinct from one another, such as the flexion of different fingers. For example, despite being some of the largest and most popular sEMG datasets, the Ninapro (Atzori et al., 2014) corpus is predominantly focused

on recognizing isolated gestures and contains only 77 subjects in its largest dataset. The EPN dataset (Benalcazar et al., 2020) is much larger but still focuses on gesture recognition, a task in which relatively simple models (LSTM) seem to saturate performance (Eddy et al., 2024), reaching 98% accuracy on cross-session generalization and 93% on cross-user generalization. Most other datasets, such as the ones from Amma et al. (2015, 5 subjects) and Ortiz-Catalan et al. (2013, 17 subjects), are even smaller and still primarily based on single-gesture/movement recognition. In contrast, an effective human-computer interface (HCI) should have the ability to disentangle multiple sequential gestures (that may overlap with each other) and generalize across a diverse set of users.

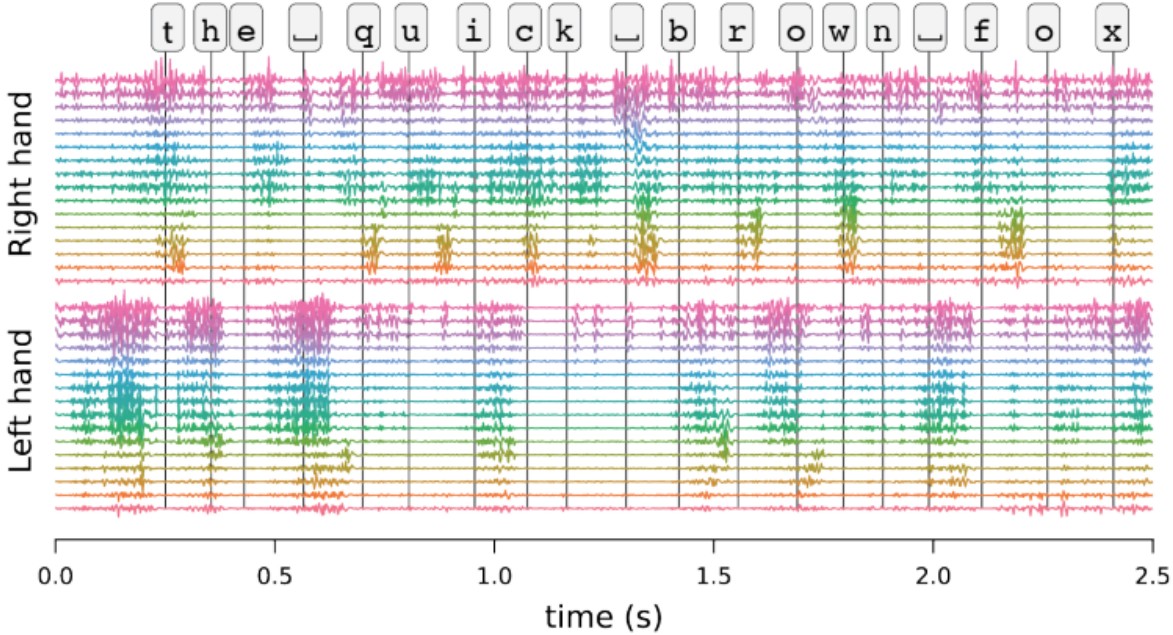

Figure 1: The *emg2qwerty* task: participants type on a keyboard while sEMG activity is recorded from both hands. The goal is to map from sequences of sEMG signals to sequences of characters. Figure cropped from `https://github.com/facebookresearch/emg2qwerty`, licensed CC BY-NC-SA.

Improving on the above, Sivakumar et al. (2024) released a dataset that represents a significant advancement over existing sEMG benchmarks in terms of its scale, task complexity, and real-world applicability. As described in Figure 1, the task is to predict key presses while touch typing on a keyboard using sEMG activity alone. The dataset captures dynamic typing behavior across 108 users and 1,135 sessions totaling 346 hours of high-resolution wrist-based sEMG recordings. The naturalistic, high-dimensional output space (key pressed on a keyboard) and the larger data scale make it suitable for studying both cross-user zero-shot generalization and personalized finetuning on unseen (during training) users. For these reasons, we focus on this dataset in our experiments.

As a result of both limited data availability and the inherent challenges of the modality (e.g., noise, subject-based variance), prior works have historically tended to utilize classical approaches for sEMG tasks. In some, simple machine learning methods such as support vector machines, random forests, K-nearest-neighbors or linear discriminant analysis are used (Rani et al., 2024; Atzori et al., 2014). In others, models such as CNNs (Ozdemir et al., 2020; Atzori et al., 2016) and LSTMs (He et al., 2018) have been applied, primarily for gesture classification on a restricted set of users. Some recent work have begun to explore the application of transformer-based models (Lai et al., 2023; Wanga et al., 2024; Dai et al., 2023; Zabihi et al., 2023; Putro et al., 2024; Chen et al., 2023; Zhang et al., 2022; 2023a;b) and vision transformers (ViTs) to sEMG data (Scheck & Schultz, 2023; Rahimian et al., 2021; Yang et al., 2024; Godoy et al., 2022; Montazerin et al., 2023; Liu et al., 2024), but these efforts are also mostly limited to small-scale datasets ($< 50$ participants,

many of them on a subset of Ninapro databases) and model sizes ($< 10M$ parameters). Additionally, many introduce significant complexity such as bespoke deep learning architectures (Wanga et al., 2024; Zabihi et al., 2023; Putro et al., 2024; Chen et al., 2023; Zhang et al., 2022; 2023a;b; Liu et al., 2024) or use manual feature extraction techniques on sliding windows of the sEMG signals, e.g., spectrograms (Sivakumar et al., 2024), multivariate power frequency features (CTRL-labs at Reality Labs, 2024), Hjorth parameters (Rani et al., 2024) and others (Zhang et al., 2022; 2023a; Putro et al., 2024). We instead focus on learning vanilla transformer-based models that use learned sEMG featurization directly from the raw data.

Finally, the existing literature often falls short in addressing a critical challenge of sEMG data: cross-user generalization. It is well-established that sEMG signals, like other bio-signals, exhibit high inter-individual variability (Chowdhury et al., 2013; CTRL-labs at Reality Labs, 2024), making it essential to evaluate the robustness of models on "unseen" (heldout during training) users. However, many prior works neglect this crucial consideration, instead opting to test their models solely on held-out *trials* from the same individuals contained within the training set. This does not provide a meaningful assessment of the model's ability to generalize across diverse users. We focus our evaluation on new participants unseen in the training set.

Table 1 reviews recent work in the domain of sEMG decoding, showing that for the most part, datasets used are small ($< 100$ participants, or even $< 10$) and modeling techniques used are often classical.

## 2.2 Knowledge distillation

Knowledge distillation was popularized by Hinton et al. (2015), who proposed training a small 'student' model on the outputs (logits) of a larger pretrained 'teacher' model, along with the ground-truth labels from the training data. This improved the performance of the smaller 'student' model compared to the case where the smaller model was trained from scratch. Subsequent work hypothesized that distillation helps because the 'teacher' model's logits provide information about interclass relationships (Tang et al., 2020) as well as sample difficulty (Zhao et al., 2022).

Other approaches go beyond using outputs or logits alone for distillation, especially for deeper models. These approaches push the 'student' model's intermediate layer representations 'close' to the intermediate layer representations in the 'teacher' model (Romero et al., 2015). In practice, this is achieved by regularizing the distance (e.g. $\ell_1$, $\ell_2$) between the activations of the 'teacher' and the 'student' model for pre-determined pairs of layers. These feature distillation methods have been successfully used to distill large foundation models like HuBERT with minimal performance degradation (Lee et al., 2022; Peng et al., 2023; Wang et al., 2023). Komodakis & Zagoruyko (2017) proposed applying a function that maps hidden representations of a CNN to a 2-D attention map and training the 'student' model to imitate the attention map from the 'teacher' instead of the features from the 'teacher'. Chen et al. (2021) eliminates the need for ad-hoc mapping functions by learning the optimal 'student-to-teacher' mapping layer. In contrast to directly minimizing some form of distance measure, Xu et al. (2018) proposed training an adversarial discriminator network to distinguish between the representations from the 'teacher' and the 'student'.

Even more sophisticated approaches encode 'teacher' knowledge in higher-order relationships between multiple samples. For example, Tung & Mori (2019) proposed computing an inner-product-based similarity matrix between a batch of samples for both the 'teacher' and 'student' models and then training the student to match the teacher's matrix. Park et al. (2019b) presented a similar approach, exploring both Euclidean and angular similarity metrics. However, the benefits of these relational distillation methods can be marginal compared to feature-based approaches, especially considering the added computational complexity.

Distillation has additionally been explored for models trained on sEMG datasets. For instance, Lai et al. (2023) distilled a ResNet model ensemble into a single model instance with good success. However, their evaluation is limited to intra-user and intra-session generalization across only five users. In contrast, our work: (i) focuses on cross-user evaluation, (ii) trains on a significantly larger dataset (346 hours compared to 10 hours) and model ($\sim$130M parameters compared to $\sim$6x6M parameter ensemble), (iii) distills the model extensively (up to 180x size reduction range compared to up to 10x range) which contributes to a better understanding of different regimes where the distillation works effectively and where it yields diminishing returns. Another example is work from Wanga et al. (2024), which modifies the transformer attention with depthwise-followed-by-pointwise convolutional feature projections, replaces the MLP part of the teacher

model with a variational information bottleneck layer (with removed skip connection) and performs simultaneous logit and feature distillation. This complexity hinders the broad applicability of the approach, which furthermore does not investigate (a) how the teacher models perform at larger scales or (b) how the distillation procedure behaves when the teacher-to-student parameter reduction is larger. In contrast, we show that the *simplest* distillation approach works and scales, making it usable for in-the-wild sEMG practitioners.

Other research works have investigated cross-modal knowledge distillation in the context of sEMG tasks. Notably, Dai et al. (2023) examined distillation between sEMG and hand gestures, while Zeng et al. (2022) explored distillation between sEMG and ultrasound. In contrast, our work concentrates on distilling knowledge from models trained exclusively on sEMG signals.

## 3  Experiments

We demonstrate that:

1. Transformer models can be effective for sEMG tasks even when considering datasets and models that are small by modern deep learning standards.

2. The performance of the transformer models further improves with scale, resulting in improvements to SOTA performance.

3. Transformer models can be distilled into smaller-sized models, recovering most of the large-model performance with 50x fewer parameters.

We primarily focus on zero-shot performance on held-out users, and additionally report personalization performance (where a model is fine-tuned on a small amount of data from a held-out single user, then evaluated on held-out sessions from that user). Both cross-user generalization and personalization are key challenges for real-world sEMG tasks (CTRL-labs at Reality Labs, 2024; Saponas et al., 2008; Yang et al., 2024).

### 3.1  Dataset

We use the *emg2qwerty* dataset (Sivakumar et al., 2024) in our experiments. The dataset consists of two-handed sEMG recordings from users typing on a computer keyboard. The data is labeled with the corresponding keystrokes, and the task is to map from sEMG sequences to character sequences. Figure 1 shows a representative example. In total, the dataset contains 346 hours of sEMG recordings across 108 unique users. The dataset is split into 100 users for training and validation and 8 held-out users for testing. For each user, we hold out 2 validation sessions and 2 testing sessions, then use the rest for training. In the generic setting, we train on the 100 user training set, validate on the 100 user validation set and evaluate on the 8 user testing set. In the personalization setting, for each of the 8 users, we train on their individual training set, then validate and test on their respective validation and testing set. Sessions are windowed to form 4 second samples, padded with an additional 900 ms of past context and 100 ms of future context.

### 3.2  Models

We compare the performance of our transformer model against the Time-Depth Separable Convolutional Network (TDS-ConvNet) model introduced in  Hannun et al. (2019) and used by Sivakumar et al. (2024), which reports that the parameter-efficiency of TDS-ConvNet allows for wider receptive fields which have proven important in *emg2qwerty* modeling.  We use the same train, validation, and test splits as used in Sivakumar et al. (2024) and report the performance of the baseline models from that paper.

Our model architecture consists of a convolutional featurizer followed by a transformer encoder and a linear decoder. The featurizer always uses 3 convolutional layers, with instance norm applied along the time axis after the first convolutional layer, and downsamples the input sEMG data (which is sampled at 2kHz) to a sequence of features (sampled at 100 Hz). The encoder consists of a series of Transformer blocks (Vaswani

et al., 2017) whose number and width we manipulate to create larger or smaller models. The transformer blocks use causal attention so that they can be used in an online streaming setup. The linear decoder converts the transformer's output to a sequence of logits. Note that unlike Sivakumar et al. (2024) which uses log-spectrograms as input features, we train our model end-to-end using raw sEMG data directly, without using any hand-designed feature engineering pipeline. Following Sivakumar et al. (2024), during training we apply channel rotation, which randomly shifts the sEMG channels by $\pm 1$ as a data augmentation technique to simulate different spatial orientations of the device.

We have trained 20 different architectures, generated by permuting [2, 4, 6, 8, 10] layers and inner dimension of [128, 256, 512, 1024]. The ratio of the transformer inner dimension and the transformer feed-forward dimension is fixed at four. While we report on performance of all models in the supplement, for ease of exposition in the main text, we are concerned with three 'reference' architectures: the TINY architecture consisting of 10 layers of inner dimension 128 (about 2.2M parameters); the SMALL architecture consisting of 6 layers of inner dimension 256 (about 5.4M parameters, close to the 5.3M TDS-ConvNet baseline); and the LARGE architecture consisting of 8 layers of inner dimension 1024 and about 109M parameters. We use the AdamW optimizer (Loshchilov, 2017) for training all models. In the figures we additionally include other models along the size-performance Pareto frontier (i.e. ones which perform better than any model of the same or lesser parameter count). For all the experiments, we report standard deviation across multiple seeds (6 seeds for supervised training of transformer models, 3 seeds for personalization experiments, and 3 seeds for distillation experiments).

### 3.2.1 Supervised training of transformer models

Following Sivakumar et al. (2024), we use the connectionist temporal classification (CTC) loss (Graves et al., 2006) to train the transformer models on the *emg2qwerty* task. We train the transformer models on a single HPC node containing 8 32GB V100 GPUs, in a Distributed Data Parallel (DDP) training scheme. We use cosine learning-rate scheduling (Loshchilov & Hutter, 2017) with linear warmup for first 5% of updates. We document all the hyperparameters in the Appendix in Section B.

### 3.2.2 Model distillation

We use logit distillation to distill the pretrained LARGE teacher model into a set of smaller student models of different sizes. For the student models, we include all 20 architectures discussed above (including ones of the same size as the teacher). This enables us to demonstrate the benefits of distillation at varying student model sizes. The process is depicted in Figure 2.

The distillation loss $\mathcal{L}_{\text{distill}}$ is the cross-entropy loss where the student's output probabilities at each time step are expected to match the soft targets provided by the teacher's per-timestep output probabilities. We use a temperature scaling factor of 2 when computing the probabilities. The distillation loss is then combined with the task loss (i.e. CTC) to give the final training loss for the student model:

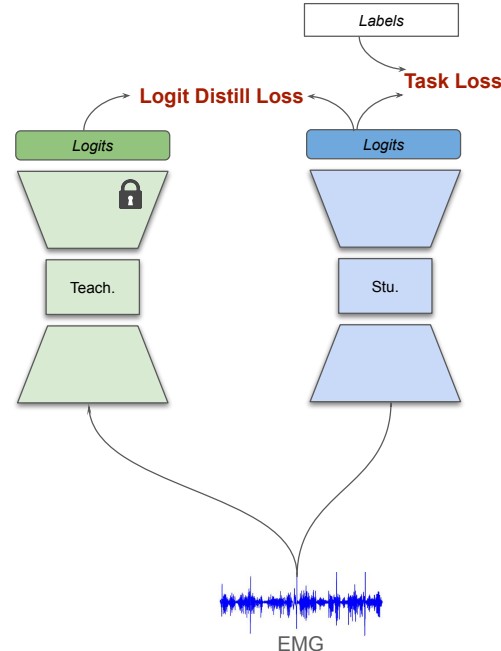

Figure 2: An illustration of the distillation process: the smaller student model receives training signal from the logits of the larger teacher model in addition to the regular supervised loss.

$$\mathcal{L} = \frac{1}{\alpha + \beta}\big(\alpha \mathcal{L}_{\text{distill}} + \beta \mathcal{L}_{\text{task}}\big). \qquad (1)$$

Here, $\alpha$ and $\beta$ are hyperparameters. We set $\beta = 1.0$ and experimented with $\alpha$ values in [0.1, 2] and found the best values to be in the [0.3, 0.5] range, we use $\alpha = 0.5$ throughout our distillation experiments. The rest of the distillation hyperparameters used in our experimentation are documented in Appendix B.3.

| Benchmark | Architecture | Parameters | CER ($\downarrow$%) |
|---|---|---|---|
| Generic | TDS-ConvNet | 5.3M | 55.57 |
| | TINY TRANSFORMER | 2.2M | 35.9± 0.9 |
| | SMALL TRANSFORMER | 5.4M | 35.2± 1.1 |
| | LARGE TRANSFORMER | 109M | 30.5± 0.6 |
| Personalized | TDS | 5.3M | 11.39 |
| | TINY TRANSFORMER | 2.2M | 9.7± 0.13 |
| | SMALL TRANSFORMER | 5.4M | 7.9± 0.06 |
| | LARGE TRANSFORMER | 109M | 6.8± 0.07 |

Table 2: Cross-user performance of transformer models trained on the *emg2qwerty* dataset, showing that [a] even the TINY transformer models substantially outperform the TDS-ConvNet baseline in spite of having fewer parameters, and [b] the performance of the transformer model keeps improving as we increase the number of parameters in the model. For the transformer models, we report standard deviation across 6 seeds for Generic benchmark and across 3 seeds for the Personalized benchmark. The standard deviation for the baseline models is not reported in Sivakumar et al. (2024).

## 3.3 Metrics

Following (Sivakumar et al., 2024), we evaluate the *emg2qwerty* models using Character Error Rate (CER), defined as the Levenshtein edit-distance between the predicted and the ground-truth sequence. It can be expressed as $\frac{(S+D+I)*100}{N}$ where, given the predicted and the ground-truth sequences, $S$ is the number of character substitutions, $D$ is the number of deletions, and $I$ is the number of insertions between them and $N$ is the total number of characters in the ground-truth sequence.

## 4 Results

### 4.1 Parameter-matched comparison of transformer with TDS baseline

In Table 2, we compare the performance of our transformer models with the baseline TDS-ConvNet model. On the generic (zero-shot, cross-user) benchmark, even the TINY model outperforms the TDS-ConvNet model baseline by a margin of about 20% absolute CER, in spite of having fewer than half the number of parameters as the baseline. This indicates that the performance gain from the transformer model arises from the specifics of the architecture (e.g., self and cross attention) and not merely from higher parameter counts. While the performance of the SMALL (which is about the same size of the baseline) is not very different from that of the TINY model, scaling the model more aggressively to the LARGE size does yield a further 5% improvement. This is surprising since common deep learning wisdom indicates that this level of increase in model capacity with a fixed dataset size would result in increased overfitting to the training users, thus hurting unseen user performance.

On the personalized benchmark, where the trained model is fine-tuned on a single heldout user's data, CERs are much lower across the board and therefore the absolute gains of the transformer are more modest (1.7%, 1.8% and 4.5% CER for TINY, SMALL and LARGE models respectively). However, the relative magnitude of improvement is similar to that seen on the generic benchmark, especially for the largest model.

These performance gains over the baseline across all model sizes and benchmark support the usage of transformer architecture for text-transcription based sEMG tasks. We compare our model architecture with other common ones from the sEMG field (Time-CNN, LSTM and Vision Transformers) in Appendix C.1.

| Benchmark | Model Architecture | Params | CER (↓%) Standard | Distilled | Abs. Gain (↑%) |
|---|---|---|---|---|---|
| Generic | TINY | 2.2M | 35.9± 0.9 | 31.9± 0.4 | 4.0 |
| | SMALL | 5.4M | 35.2± 1.1 | 32.7± 0.5 | 2.5 |
| Personalization | TINY | 2.2M | 9.7± 0.1 | 8.6± 0.04 | 1.1 |
| | SMALL | 5.4M | 7.9± 0.06 | 7.1± 0.06 | 0.8 |

Table 3: Cross-user performance of small student models on the *emg2qwerty* dataset with and without distillation. Performance is measured by character error rate (CER). The 'Abs. Gain' column reflects the absolute improvement in performance from using distillation as opposed to standard supervised training for a given architecture. Personalized models are personalized from the distilled student. All models see a substantial benefit (7-11% relative improvement) from the use of the distillation loss. We report standard deviation across 3 seeds for the distillation results. Here TINY and SMALL refer to TINY TRANSFORMER and SMALL TRANSFORMER in Table 2 respectively.

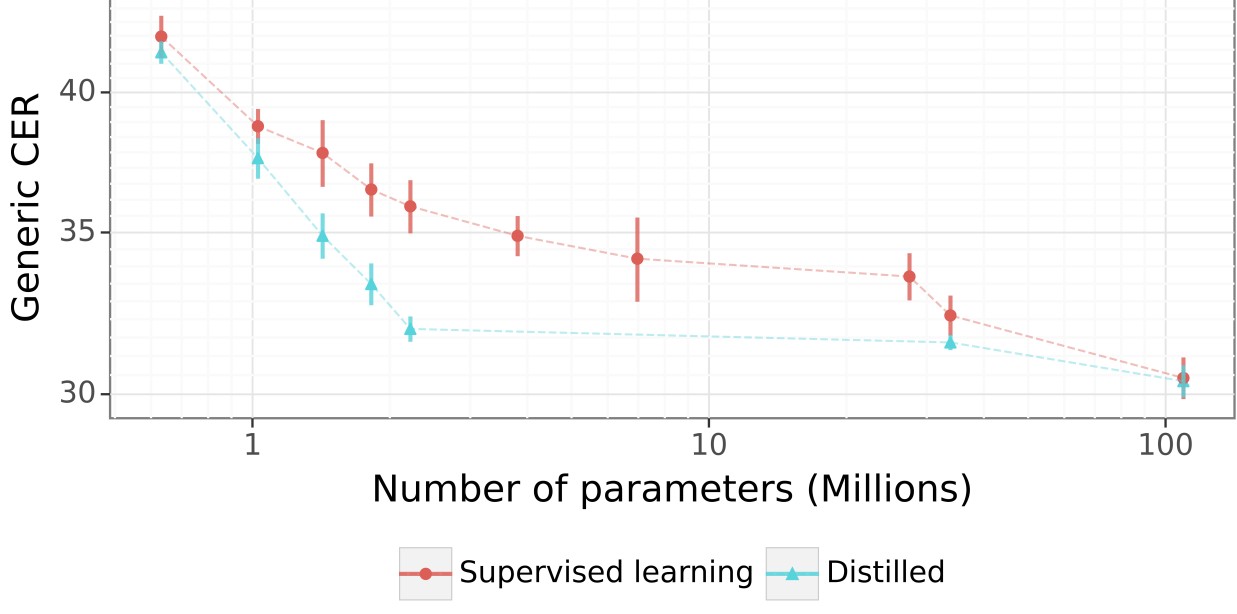

Figure 3: Scaling curve of transformers on the *emg2qwerty* dataset, showing the benefit of model size across 3 orders of magnitude from <1M to over 100M parameters, and the benefit of distillation. A few things are of note: [a] even the smallest transformer we consider here (about 600K parameters) outperforms the 5.3M parameter TDS-ConvNet baseline (55.57 CER), in spite of having almost an order of magnitude fewer parameters; and [b] the majority of distillation benefit is seen for small-but-not-too-small models, with a 2.2M parameter distilled model getting within 1.5% CER (absolute) of the top-performing model in spite of having 50x fewer parameters. Figure includes only models along the pareto front, i.e. ones which outperform others at the same or smaller parameter count. The vertical bars denote standard deviation across 6 seeds.

## 4.2 The performance of transformer models improves with scale

In Table 2, we observe that as we scale the size of the transformer model, the performance of the model continues to improve even with the same amount of data. An alternate view of this is shown in Figure 3 (red line) where we report on the 10 architectures that are on the Pareto front w.r.t. scale and performance,

i.e. they perform at least as well as any model of the same or smaller size. These models show a nearly log-linear scaling curve, validating that there are performance benefits to be realized by using larger transformer models for sEMG data.

One obvious challenge with the use of these larger transformer models is that they increase the computational overhead during inference, thus limiting the potential for real-time usage in an HCI (e.g. to control a prosthetic, for computer text entry, or to control a cursor). Distillation techniques mitigate this limitation, which we investigate in the following section.

### 4.3 Model distillation can be applied effectively to models for sEMG

Table 3 shows the results of distilling small-capacity student models on the *emg2qwerty* dataset, using the best-performing LARGE model as a teacher. An alternate view is in Figure 3 (turquoise line), where we see that the benefit of distillation is largest for small-but-not-too-small models. That is, for the very tiniest models, they seem unable to fully benefit from distillation (perhaps too few layers, or too few parameters to model the teacher's distribution), while for bigger models they have enough capacity to learn the task on their own. Notably, the 2.2M parameter TINY achieves performance within 1.5% of the LARGE teacher in spite of having about 50x fewer parameters. We also report that while the LARGE teacher model takes 26.966 ms for inference over a single sample, the SMALL model takes 5.7002 ms only, thus providing a $4.7\times$ speedup during inference. These numbers are obtained by running inference over a single sample 1000 times to get an average runtime and then further repeating this protocol 3 times and reporting the median of the three numbers. Additional details around benchmarking are provided in the Appendix. While the specific size cutoff for a real-time model may vary, distillation provided a benefit across many small model sizes. These result indicate that, for student models within the appropriate size range, the benefits of distillation are substantial in maximizing performance for a given model size. Obtaining such large (50x) parameter reductions at relatively minimal performance cost represents an important step towards models that can be used for practical BCIs outside of clinical or laboratory environments.

## 5 Analysis

In Section 4.1, we report considerable improvement of our modeling approach over results reported in Sivakumar et al. (2024). In this section we investigate which differences contributed to these performance gains. We identify and ablate two core differences between our approach and Sivakumar et al. (2024): first the featurizer and second the encoder type. Additionally, we investigate the contribution of data augmentation techniques to the training procedure. The technical details of these experiments are outlined in Appendix B.5.

**Featurizer type**  We refer to the "featurizer" as all modeling parts between the raw sEMG signal and the core encoder architecture, i.e., the Transformer in our approach or the TDS model in the case of Sivakumar et al. (2024). A featurizer is commonly employed in the deep learning literature as a way to create meaningful vector representations of the data for the encoder to process. In the case of sEMG, part of this featurization procedure has historically often been handcrafted using transformations of the data that are considered useful for the task, (e.g. Mean Absolute Value (MAV), Variance (VAR), Root Mean Square (RMS), spectrograms, etc). In this work, we adopt a data-driven approach to featurization through a CNN architecture applied directly to the raw sEMG signal, as popularized by Schneider et al. (2019). We perform the ablation experiment between our raw sEMG signal+CNN→encoder featurization strategy to the spectrogram+MLP→encoder strategy from Sivakumar et al. (2024) with the TDS and Transformer encoders in Table 4. We see that TDS and Transformer encoders benefit 10.94 and 8.19 absolute CER improvement, respectively, from using a data-driven featurization approach. In other words, empirically derived features are better able to capture and encode the low-level information required for the emg2qwerty task. Importantly, this is true not only for Transformer encoders, but even for the baseline TDS encoder.

**Architecture ablation**  In addition to the featurization strategy, we investigate the contribution of using a transformer encoder instead of a TDS encoder. In Table 4, we show the performance of each encoder type. We see that the transformer outperforms the TDS encoder by 8.88 and 11.63 CER, depending on the

| | | Featurizer type | |
|---|---|---|---|
| | | raw sEMG signal+CNN | Spectrogram+MLP |
| **Encoder type** | TDS | 45.84± 0.06 | 56.78± 0.17 |
| | Transformer | 36.96± 0.95 | 45.15± 1.17 |

Table 4: Featurizer and Encoder type ablation on cross-user performance when trained from scratch without distillation signal. Performance is measured by character error rate (CER) on the *emg2qwerty* dataset. We report standard deviation across 3 seeds.

| | | Input masking | | |
|---|---|---|---|---|
| | | None | Length=7 | Length=15 |
| **Channel rotation** | None | 36.24± 1.04 | 33.96± 1.27 | 33.05± 0.24 |
| | ± 1 | 36.92± 0.82 | 33.32± 0.97 | 34.77± 1.59 |

Table 5: Data augmentation ablation on cross-user performance when trained from scratch without distillation signal. We investigate the cross-product of 3 input masking and 2 channel rotation configurations. Performance is measured by character error rate (CER) on the *emg2qwerty* dataset. We report standard deviation across 3 seeds.

featurizer type. We hypothesize that the transformer is better able to capture the sequential nature of the typing signal by encoding information on both local and global timescales. The attention mechanism in the transformer may also enhance the model's ability to discard noise and other irrelevant features, a known challenge in sEMG data (Chowdhury et al., 2013; Cochrane-Snyman et al., 2016).

**Data augmentation ablation**   We also investigate contribution of data augmentation techniques used in this work, specifically input masking, also called SpecAugment (Park et al., 2019a) as used by Sivakumar et al. (2024)), and input channel rotation. For both of these augmentations, we evaluate the Transformer with raw sEMG signal+Conv input feature type shown in Table 4 and show the results in Table 5. We see that using input masking improves performance over not using the augmentation; however, there does not seem to be a clear optimal masking length between 7 and 15. Adding channel rotation augmentation doesn't appear to provide substantive performance improvements over not using it. We see an exception to that when using input masking with mask lengths of 15, in which case not having rotation augmentation improves performance by approximately 1.7 CER, indicating a potential interaction between the two augmentation techniques.

## 6   Conclusion

Our work provides a new state-of-the-art on the *emg2qwerty* task without increasing the parameter count over the baseline. We achieved this with a simple and pragmatic recipe that can easily be followed by practitioners in other task settings. First, we scale up a vanilla Transformer architecture, which yields improved performance at all sizes investigated in this work (up to 110M parameters). Second, if bigger models cannot be deployed because of compute or other constraints, we show that the simplest form of knowledge distillation works to bring scaled models back into a usable regime (up to 50x reduction in size) with minimal performance degradation. Third, we provide an empirical limit where this approach seems to yield diminishing returns (>50x size reduction). We perform all of the evaluations in the challenging cross-user generalization setting. We see our work as providing a pragmatic recipe applying standard and battle-tested tools to deep learning for sEMG at scale.

**Broader Impact Statement**

Beyond this paper, the broader usage of sEMG, and the specific development sEMG-based textual input models, pose novel ethical and societal considerations. There are numerous societal benefits for the development of sEMG models for textual input. sEMG allows one to directly interface a person's neuromotor intent with a computing device, which can be used to, for example, develop adaptive controllers for those who struggle to use existing computer interfaces.

**Acknowledgments**

Thank you to Paul Parayil Varkey, Ishita Mediratta, Brett Nelson, Roman Rädle, Daniel Licht, and Andy Chung for extensive contributions to related code and research that directly helped to enable this work. For discussions on techniques for modeling sEMG, thank you to Alex Gramfort, Eftychios Pnevmatikakis, Nariman Farsad, Ali Farschian, Na Young Jun, Rick Warren, Tiberiu Tesileanu, Adam Calhoun, and Alan Du. Thanks to Viswanath Sivakumar for emg2qwerty support, and to Viswanath, Dan Wetmore, and Patrick Kaifosh for reviewing the paper. Thank you to Michael Mandel and Sean Bittner for reviewing the code.

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

## Appendix

The appendix is organized as follows: In Section B we lay out all of the experimental details of the results presented in this work and in Section C we show additional results which were not presented in the main text.

## A    Extended related works

Table 6: Overview of related works to this paper along our contribution axes. For the model axis, we distinguish 4 subcategories: Architecture, "Vanilla" (whether it the architecture is used as-is or modified specifically for sEMG), Raw features (whether the model processes sEMG signal of a handcrafted representation of it) and the number of parameters. For the distillation axis, we distinguish between 3 subcategories: whether it does distillation at all, "Vanilla" (again, used as-is or modified specifically for sEMG) and the model size reduction achieved. Along the evaluation axis, we distinguish between 2 subcategories: Whether is it evaluating cross-user generalization and whether the core evaluation of the approach was done on Ninapro databases.

| Paper | Arch | Model Vanilla | Raw Features | #params | Does KD? | Distillation Vanilla | Size reduction | Evaluation Cross-user? | Is Ninapro? |
|---|---|---|---|---|---|---|---|---|---|
| Lai et al. (2023) | Resnet | YES | YES | 35M (ensemble)[a] | YES | YES | [4-10]x | NO | NO |
| Wanga et al. (2024) | Transformer | NO | YES | <1M | YES | NO | ∼7x | NO | YES |
| Dai et al. (2023) | Transformer | NO | YES | Not specified | YES | NO | NA | NO | YES |
| Zeng et al. (2022) | CNN | NO | YES | Not specified | YES | NO | 2x | NO | YES |
| Rahimian et al. (2021) | ViT | YES | YES | <100k | NO | | | NO | YES |
| Yang et al. (2024) | ViT | YES | YES | 6M | NO | | | YES | YES+others |
| Scheck & Schultz (2023) | Transformer | YES | YES | Not specified | NO | | | YES | NO |
| Godoy et al. (2022) | ViT | YES | YES | Not specified | NO | | | NO | YES |
| Montazerin et al. (2023) | ViT | YES | YES | <500k | NO | | | NO | NO |
| Zabihi et al. (2023) | Transformer | NO | YES | <1M | NO | | | NO | YES |
| Putro et al. (2024) | Transformer | NO | NO | Not specified | NO | | | NO | YES |
| Chen et al. (2023) | Transformer | NO | YES | Not specified | NO | | | NO | NO |
| Zhang et al. (2022) | Transformer | NO | NO | Not specified | NO | | | NO | YES |
| Zhang et al. (2023a) | Transformer | NO | NO | <500k | NO | | | NO | YES |
| Zhang et al. (2023b) | Transformer | NO | YES | Not specified | NO | | | NO | YES+others |
| Liu et al. (2024) | ViT | NO | YES | Not specified | NO | | | NO | YES+others |

[a] The model size reduction is not specified by Lai et al. (2023), we infer an approximation of the size reduction from the model memory footprint (assuming FP32 weights).

## B    Experimental details

We describe the hyperparameters and model details in the following subsections. We provide constant hyperparameters in one table and hyperparameters that are subject to vary in a separate table. In Section B.2, we list the hyperparameters of the supervised learning models presented in this paper, i.e., without distillation signal; in Section B.3, we list the hyperparameters of the distilled models; and in Section B.4 we describe the hyperparameters of the personalization experiment.

### B.1    Dataset details

In Table 7, we detail the aggregated statistics of the *emg2qwerty* dataset.

### B.2    Supervised learning details

In the supervised learning experiments, we train a grid of model with sizes $[128, 256, 512, 1024]$ transformer hidden representation size and $[2, 4, 6, 8, 10]$ layers. For each of the hidden representation sizes, we set the transformer feed-forward dimension such that the feed-forward ratio ($d_{\text{ff}}/d_{\text{hidden}}$) is maintained at 4. For each of these configuration, we launch multiple learning rates ($[3e-3, 1e-3, 3e-4, 1e-4]$) across 6 different seeds. Seeding is used to determine dataloading order and model initialization. We train all models to completion (200 epochs) and evaluate the model on the training and validation at the end of every epoch. For each model, we artificially early-stop the model post-hoc by choosing the epoch with the lowest validation CER and record the test set CER of the model for this epoch. The rest of the training-related hyperparameters

Table 7: *emg2qwerty* dataset statistics

| | |
|---|---|
| Total subjects | 108 |
| Total sessions | $1,135$ |
| Avg sessions per subject | 10 |
| Max sessions per subject | 18 |
| Min sessions per subject | 1 |
| Total duration | 346.4 hours |
| Avg duration per subject | 3.2 hours |
| Max duration per subject | 6.5 hours |
| Min duration per subject | 15.3 minutes |
| Avg duration per session | 18.0 minutes |
| Max duration per session | 47.5 minutes |
| Min duration per session | 9.5 minutes |
| Avg typing rate per subject | 265 keys/min |
| Max typing rate per subject | 439 keys/min |
| Min typing rate per subject | 130 keys/min |
| Total keystrokes | $5,262,671$ |

(specifically dropout probabilities, weight decay and learning rate schedule) were chosen according to best values in prior experimentation with a fixed model size.

We aggregate the results by first taking the validation and test CER averages across seeds, then pick the best aggregated validation learning rate value and report the average and standard deviation of the test results. The chosen hyperparameters are reported in Table 9 and the test CER reported in Table 21.

### B.3 Distillation details

In the distillation experiments, we train the same grid of model width and depth as in the supervised learning experiments (Section B.2) to use as the student model. The teacher model is chosen by picking the best performing model from the best model configuration from the supervised learning experiments in Table 21. For each of the hidden representation sizes of the student model, we set the transformer feed-forward dimension such that the feed-forward ratio ($d_{\text{ff}}/d_{\text{hidden}}$) is maintained at 4. For each of these configurations, we launch multiple learning rates ($[3e-3, 1e-3, 3e-4]$) across 3 different seeds. Seeding is used to determine dataloading order and model weight initialization. We train all models to completion (200 epochs) and evaluate the model on the training and validation at the end of every epoch. For each model, we select the checkpoint with the lowest validation CER and record the test set CER of this checkpoint. The rest of the training-related hyperparameters (specifically distillation penalty weight, student dropout probabilities, weight decay and learning rate schedule) were chosen according to best validation metrics in prior experimentation with a fixed model size.

We aggregate the results by first taking the validation and test CER averages across seeds, then pick the best aggregated validation learning rate value and report the average and standard deviation on the test results. The chosen hyperparameters are reported in Table 11 and the test CER reported in Table 22.

### B.4 Personalization details

In the personalization experiments, we focus on our three highlighted architecture configurations, i.e., TINY, SMALL and LARGE. For each architecture, we pick three models from the supervised learning experiments, irrespective of hyperparameters and seeds, according to the best validation performance. We refer to these three models as seed A, B and C. For the TINY and SMALL architecture, we repeat this procedure with the distillation set of experiments. We refer to these models as being from the "Distillation" origin, as opposed

Table 8: Supervised learning task details and hyperparameters held constant for models from Table 21.

| | | |
|---|---|---|
| Data | Input sEMG channels | 32 |
| | Window length | 8000 |
| | Padding | $[1800, 200]$ |
| Architecture | Featurizer channels | $[128, 64, 64]$ |
| | Featurizer kernels | $[11, 3, 3]$ |
| | Featurizer strides | $[5, 2, 2]$ |
| | Encoder feed-forward ratio | 4 |
| | Encoder attentions heads | 16 |
| | Tokenizer | Character-level |
| | Vocab size | 99 |
| | Encoder hidden, attention and activation dropout | 0.2 |
| | Encoder feature projection dropout | 0.2 |
| | Encoder final layer dropout | 0.2 |
| Training | Epochs | 200.0 |
| | Effective batch size | 640 |
| | Encoder causal attention | True |
| Optimizer | Learning rate schedule | linear warmup + cosine decay |
| | Learning rate warmup ratio | 0.05 |
| | Weight decay | 0.2 |
| | CTC zero infinity | True |
| | Gradient clipping | 0.1 |
| Software | Torch version | 2.3.1+cu121 |
| | Transformers version | 4.36.2 |

to the "Supervised" origin for the models taken from the supervised learning experiments. For each of those models, we initialize the personalization models from the chosen checkpoint and we launch multiple learning rates ($[3e-3, 1e-3, 3e-4]$) across 3 different seeds for each of the 8 personalization users. Seeding is used to determine dataloading order and model weight initialization. We train all models to completion (100 epochs) and evaluate the model on the training and validation sets at the end of every epoch. For each model, we select the checkpoint with the lowest validation CER and record the test set CER of this checkpoint. The rest of the training-related hyperparameters (specifically student dropout probabilities, weight decay and learning rate schedule) were chosen according to best validation metrics in prior experimentation with a fixed model size.

We aggregate the results by first taking the validation and test CER averages across the personalized users and seeds. We then pick the best aggregated validation learning rate value and report the average and standard deviation on the test results. The chosen hyperparameters are reported in Table 13 and the test CER reported in Table 23. Table 13 also reports the supervised or distillation learning rate used by the seed generic model the personalized was initialized from.

## B.5 Ablation experiments

In the ablation experiments, we focus on a single architecture configurations for each investigated variations, i.e., Raw sEMG/spectrogram, Transformer/TDS, With/without channel rotation and with/without input masking. All of the configuration for these experiments can be found in Table 14 for the architecture ablation and Table 16 for augmentation. For each of these configuration, we launch two learning rates ($[1e-3, 3e-4]$) across 3 different seeds. Seeding is used to determine dataloading order and model weight initialization. We train all models to completion (number of epochs mentioned in the tables) and evaluate the model on the training and validation sets at the end of every epoch. For each model, we select the checkpoint with the lowest validation CER and record the test set CER of this checkpoint. The rest of the training-related

Table 9: Supervised learning task hyperparameters that vary for models from Table 21.

| Architecture | Encoder hidden size | Encoder layers | Optimizer learning rate |
|---|---|---|---|
| | 128 | 2 | 1e-03 |
| | 128 | 4 | 1e-03 |
| | 128 | 6 | 1e-03 |
| | 128 | 8 | 1e-03 |
| **Tiny** | **128** | **10** | **1e-03** |
| | 256 | 2 | 1e-03 |
| | 256 | 4 | 1e-03 |
| **Small** | **256** | **6** | **1e-03** |
| | 256 | 8 | 1e-03 |
| | 256 | 10 | 3e-04 |
| | 512 | 2 | 1e-03 |
| | 512 | 4 | 3e-04 |
| | 512 | 6 | 3e-04 |
| | 512 | 8 | 3e-04 |
| | 512 | 10 | 3e-04 |
| | 1024 | 2 | 1e-03 |
| | 1024 | 4 | 3e-04 |
| | 1024 | 6 | 3e-04 |
| **Large** | **1024** | **8** | **3e-04** |
| | 1024 | 10 | 3e-04 |

hyperparameters (specifically student dropout probabilities, weight decay and learning rate schedule) were chosen according to best validation metrics in prior experimentation with a fixed model size.

We aggregate the results by first taking the validation and test CER averages across seeds, then pick the best aggregated validation learning rate value and report the average and standard deviation of the test results. The chosen hyperparameters are reported in Table 15 and 17, while the test CERs are reported in Table 4 and 5.

## C   Additional results

In Section C.1, we show a comparison with the transformer architecture used in this work with other common architectures in the sEMG field. In Section C.2, we show training curve samples from our experiments. In Section C.3, we show the results for the full model scaling grid we explored. In Section C.4, we present measures of the inference speed of our models.

### C.1   Additional baselines

In this section, we compare the TRANSFORMER architecture used throughout this work other architecture common in the sEMG literature. We investigate three models: a Time Convolutional Neural Network (CNN), a Long Short-Term Memory (LSTM) network, and a Visual Transformer (ViT). All architectures are constructed as to have about the same number of parameters as the TDS-ConvNet baseline from Sivakumar et al. (2024). We provide high level descriptions of each in the following paragraphs, to get more details see Table 19 and Table 20.

Time-CNN : This model uses a raw sEMG+CNN featurizer (same as our TRANSFORMER architecture) and time-wise convolutional layers as encoder, each with a kernel size of 5 and a stride of 1 with channel dimensions of [512,512,512,256]. Decoding is done with a linear layer to make predictions over the dictionary.

Table 10: Distillation task details and hyperparameters held constant for models from Table 22.

| | | |
|---|---|---|
| Data | Input sEMG channels | 32 |
| | Window length | 8000 |
| | Padding | [1800, 200] |
| Student arch. | Featurizer channels | [128, 64, 64] |
| | Featurizer kernels | [11, 3, 3] |
| | Featurizer strides | [5, 2, 2] |
| | Encoder attentions heads | 16 |
| | Text Tokenizer | Character-level |
| | Vocab size | 99 |
| | Encoder hidden, attention and activation dropout | 0.2 |
| | Encoder feature projection dropout | 0.2 |
| | Encoder final layer dropout | 0.2 |
| Teacher arch. | Featurizer channels | [128, 64, 64] |
| | Featurizer kernels | [11, 3, 3] |
| | Featurizer strides | [5, 2, 2] |
| | Encoder hidden size | 1024 |
| | Encoder feed-forward ratio | 4 |
| | Encoder layers | 8 |
| | Encoder convolutional dim | [64] |
| | Encoder attentions heads | 16 |
| | Text Tokenizer | Character-level |
| | Vocab size | 99 |
| | Encoder hidden, attention and activation dropout | 0.0 |
| | Encoder feature projection dropout | 0.0 |
| | Encoder final layer dropout | 0.0 |
| Training | Epochs | 200 |
| | Effective batch size | 640 |
| | Encoder causal attention | True |
| Optimizer | Learning rate schedule | linear warmup + cosine decay |
| | Learning rate warmup ratio | 0.05 |
| | Weight decay | 0.1 |
| | CTC zero infinity | True |
| | Gradient clipping | 0.1 |
| | Distillation loss weight | 0.5 |
| | Distillation loss (logits) | Cross Entropy |
| Software | Torch version | 2.3.1+cu121 |
| | Transformers version | 4.36.2 |

LSTM : This model uses a raw sEMG+CNN featurizer (same as our TRANSFORMER architecture) and a uni-directional LSTM network as encoder with 10 layers and dimension of 256. Decoding is done with a linear layer to make predictions over the dictionary.

ViT : This model employs a transformer encoder architecture (same as our TRANSFORMER architecture). Similar to the TRANSFORMER architecture, the featurization is CNN-based applied on raw sEMG signal, but it uses non-overlapping 10ms patches that include all EMG channels. We use a 8 layer encoder, and we do a simple sweep across the dimension of the transformer to peek into the scalability of the model. Decoding is done with a linear layer to make predictions over the dictionary.

We show the results of these experiments in Table 18.

Table 11: Distillation hyperparameters that vary for models from Table 22.

| Student arch. | Student encoder hidden size | Student encoder layers | Optimizer learning rate |
|---|---|---|---|
| | 128 | 2 | 1e-03 |
| | 128 | 4 | 1e-03 |
| | 128 | 6 | 1e-03 |
| | 128 | 8 | 1e-03 |
| Tiny | 128 | 10 | 1e-03 |
| | 256 | 2 | 1e-03 |
| | 256 | 4 | 1e-03 |
| Small | 256 | 6 | 1e-03 |
| | 256 | 8 | 1e-03 |
| | 256 | 10 | 1e-03 |
| | 512 | 2 | 3e-03 |
| | 512 | 4 | 1e-03 |
| | 512 | 6 | 1e-03 |
| | 512 | 8 | 1e-03 |
| | 512 | 10 | 3e-04 |
| | 1024 | 2 | 1e-03 |
| | 1024 | 4 | 1e-03 |
| | 1024 | 6 | 3e-04 |
| Large | 1024 | 8 | 3e-04 |
| | 1024 | 10 | 3e-04 |

In Table 1, we see that the transformer model demonstrates similar performance to Vision Transformer (ViT) architecture. In contrast, the Time-CNN model exhibits significantly poorer performance compared to the other architectures. We attribute this to the Time-CNN's finite (and short) receptive field, which limits its ability to utilize past context from the sequence for predictions. Although TDS-ConvNet is also based on CNN layers, it is less affected by this limitation due to its design, which is tailored to maintain larger receptive fields (Hannun et al., 2019).

The LSTM (with Raw sEMG+CNN featurizer) model performs reasonably well, surpassing the previous TDS-ConvNet baseline (with Spectrogram+MLP featurizer) from Sivakumar et al. (2024). However, this improvement may largely be due to the enhanced featurizer (Raw sEMG+CNN vs Spectrogram+MLP). From the ablation experiments in Table 4 of the Section 5, we see that a Raw sEMG+CNN featurizer with TDS encoder performs about the same as the Raw sEMG+CNN featurizer with LSTM encoder model investigated here (i.e., about 45 CER).

## C.2 Sample training metrics

We show training curves from some of our experiments assist others to more easily reproduce our results. In Figure 4 and 5, we show training loss along with validation (cross-session) and test (cross-user) generic CER for the supervised and distillation tasks respectively. In Figure 6, we show personalization training loss and personalized CER for three of the eight personalization users.

## C.3 Full model grid scaling results

**Model architecture exploration** In Figure 7, we show the results for all model shapes investigated in this work. We use this grid of shapes to compute the pareto front, i.e. ones which outperform others at the same or smaller parameter count, presented in Figure 3. Our grid extends from an transformer hidden representation size of 128 to 1024 and a number of transformer layer from 2 to 10. The featurization module which first converts raw EMG into features to feed into the transformer is kept fixed for all models (see Table 8 and Table 10 for the exact configuration). The CER results for generic models (from Figure 7) along

Table 12: Personalization task details and hyperparameters held constant for models from Table 23.

| | | |
|---|---|---|
| Data | Input sEMG channels | 32 |
| | Window length | 8000 |
| | Padding | [1800, 200] |
| Architecture | Featurizer channels | [128, 64, 64] |
| | Featurizer kernels | [11, 3, 3] |
| | Featurizer strides | [5, 2, 2] |
| | Encoder feed-forward ratio | 4 |
| | Encoder attentions heads | 16 |
| | Tokenizer | Character-level |
| | Vocab size | 99 |
| | Encoder hidden, attention and activation dropout | 0.2 |
| | Encoder feature projection dropout | 0.2 |
| | Encoder final layer dropout | 0.2 |
| Training | Epochs | 100.0 |
| | Effective batch size | 640 |
| | Encoder causal attention | True |
| Optimizer | Learning rate schedule | linear warmup + cosine decay |
| | Learning rate warmup ratio | 0.05 |
| | Weight decay | 0.0 |
| | Training warmup ratio | 0.05 |
| | CTC zero infinity | True |
| | Gradient clipping | 0.1 |
| Software | Torch version | 2.3.1+cu121 |
| | Transformers version | 4.36.2 |

Table 13: Personalization hyperparameters that vary for models from Table 23.

| Architecture | Init. origin | Init. seed | Generic training lr | Personalization training lr |
|---|---|---|---|---|
| Tiny | Distillation | seed A | 1e-03 | 3e-04 |
| Tiny | Distillation | seed B | 1e-03 | 3e-04 |
| Tiny | Distillation | seed C | 1e-03 | 3e-04 |
| Tiny | Supervised | seed A | 1e-03 | 3e-04 |
| Tiny | Supervised | seed B | 1e-03 | 3e-04 |
| Tiny | Supervised | seed C | 1e-03 | 3e-04 |
| Small | Distillation | seed A | 1e-03 | 3e-04 |
| Small | Distillation | seed B | 1e-03 | 3e-04 |
| Small | Distillation | seed C | 1e-03 | 3e-04 |
| Small | Supervised | seed A | 1e-03 | 3e-04 |
| Small | Supervised | seed B | 3e-04 | 3e-04 |
| Small | Supervised | seed C | 3e-04 | 3e-04 |
| Large | Supervised | seed A | 3e-04 | 3e-05 |
| Large | Supervised | seed B | 3e-04 | 1e-04 |
| Large | Supervised | seed C | 3e-04 | 1e-04 |

with their standard deviation can be found in Table 21 for supervised learning and Table 22 for distilled models.

**Distillation improvement** In Figure 8, we show the generic CER improvement observed through distillation across transformer model width and depth.

Table 14: Constant hyperpameter for the featurizer and encoder architecture ablation experiment in Table 4.

| | | |
|---|---|---|
| Data | Input sEMG channels | 32 |
| | Window length | 8000 |
| | Padding | $[1800, 200]$ |
| Raw sEMG+CNN featurizer | CNN channels | $[128, 256, 384]$ |
| | CNN kernels | $[11, 3, 3]$ |
| | CNN strides | $[5, 2, 2]$ |
| Spectrogram+MLP featurizer | MLP dims | $[384]$ |
| Transformer encoder | feed-forward ratio | 4 |
| | Num. layers | 8 |
| | hidden size | 384 |
| | attentions heads | 12 |
| | hidden, attention and activation dropout | 0.2 |
| | feature projection dropout | 0.2 |
| | final layer dropout | 0.2 |
| TDS encoder | Encoder block channels | $[24, 24, 24, 24]$ |
| | Encoder kernel width | 32 |
| Task | Tokenizer | Character-level |
| | Vocab size | 99 |
| Training | Epochs | 200.0 |
| | Effective batch size | 640 |
| | Encoder causal attention | True |
| Optimizer | Learning rate schedule | linear warmup + cosine decay |
| | Learning rate warmup ratio | 0.05 |
| | Weight decay | 0.2 |
| | CTC zero infinity | True |
| | Gradient clipping | 0.1 |
| Software | Torch version | 2.3.1+cu121 |
| | Transformers version | 4.36.2 |

Table 15: Varying hyperpameter for the featurizer and encoder architecture ablation experiment in Table 4.

| Featurizer | Encoder | Optimizer learning rate |
|---|---|---|
| Raw sEMG+CNN featurizer | Transformer | 0.001 |
| Spectrogram+MLP featurizer | Transformer | 0.001 |
| Raw sEMG+CNN featurizer | TDS | 0.0003 |
| Spectrogram+MLP featurizer | TDS | 0.001 |

**Numerical results** Following is the full set of numerical performance reported in this paper. In Table 21 and 22 we annotate the TINY, SMALL and LARGE architecture which corresponds to the architecture highlighted in Table 3 of the main paper.

## C.4 Inference speed

We measured the inference speed of the highlighted TINY, SMALL and LARGE in Table 2 model architecture to assess their viability to be run in real time and to see how much model scale influences inference speeds. We present the inference speeds in Table 24.

Table 16: Hyperpameter for data augmentation ablation experiment in Table 5.

| | | |
|---|---|---|
| Data | Input sEMG channels | 32 |
| | Window length | 8000 |
| | Padding | $[1800, 200]$ |
| Architecture | Featurizer channels | $[128, 256, 256]$ |
| | Featurizer kernels | $[11, 3, 3]$ |
| | Featurizer strides | $[5, 2, 2]$ |
| | Encoder feed-forward ratio | 4 |
| | Num. Encoder layers | 8 |
| | Encoder hidden size | 256 |
| | Encoder attentions heads | 8 |
| | Tokenizer | Character-level |
| | Vocab size | 99 |
| | Encoder hidden, attention and activation dropout | 0.2 |
| | Encoder feature projection dropout | 0.2 |
| | Encoder final layer dropout | 0.2 |
| Training | Epochs | 200.0 |
| | Effective batch size | 640 |
| | Encoder causal attention | True |
| Optimizer | Learning rate schedule | linear warmup + cosine decay |
| | Learning rate warmup ratio | 0.05 |
| | Weight decay | 0.2 |
| | CTC zero infinity | True |
| | Gradient clipping | 0.1 |
| Software | Torch version | 2.3.1+cu121 |
| | Transformers version | 4.36.2 |

Table 17: Varying hyperpameter for the featurizer and encoder architecture ablation experiment in Table 5.

| Channel rotation | Input masking | Optimizer learning rate |
|---|---|---|
| None | None | 0.001 |
| ± 1 | None | 0.001 |
| None | Length=15 | 0.001 |
| ± 1 | Length=15 | 0.001 |
| None | Length=7 | 0.001 |
| ± 1 | Length=7 | 0.001 |

**Naive streaming inference** In the most naive case of streaming inference, we pass a full window length of data (4 second in emg2qwerty) to the model and use exclusively the final prediction, then repeat this process for the next prediction.

Applying this naive streaming inference paradigm to single-window inference speed in Table 24 tells us that the LARGE architecture could not be inferable in real-time due to the maximum inference frequency being approximately $f = \frac{1}{27\text{ms}} = 37\text{Hz}$.

Note that under this setting one can trade off latency with inference speed, for example by using the last $n$ predictions from a window as predictions, which alleviates the requirements on inference speed by $n$x at the cost of increasing latency of the output by $n$x.

| Architecture | Parameters | CER ($\downarrow$%) |
|---|---|---|
| TDS-ConvNet | 5.3M | 55.57 |
| Time-CNN | 5.9M | 70.14 $\pm$0.35 |
| LSTM | 5.3M | 45.96 $\pm$1.02 |
| ViT (d=128) | 1.8M | 37.29 $\pm$1.6 |
| ViT (d=256) | 6.9M | 35.29 $\pm$0.7 |
| ViT (d=512) | 27M | 34.73 $\pm$1.0 |
| TINY TRANSFORMER | 2.2M | 35.9 $\pm$0.9 |
| SMALL TRANSFORMER | 5.4M | 35.2 $\pm$1.1 |
| LARGE TRANSFORMER | 109M | 30.5 $\pm$0.6 |

Table 18: Cross-user generic performance of common architectural baselines on the *emg2qwerty* dataset. The baselines are compared to the TDS model from Sivakumar et al. (2024) and the TRANSFORMER model from the rest of our analysis. We report the standard deviation across 3 seeds for the Simple CNN, LSTM and ViT baselines, and 6 seeds for the TRANSFORMER architecture. The standard deviation for the baseline models is not reported in Sivakumar et al. (2024).

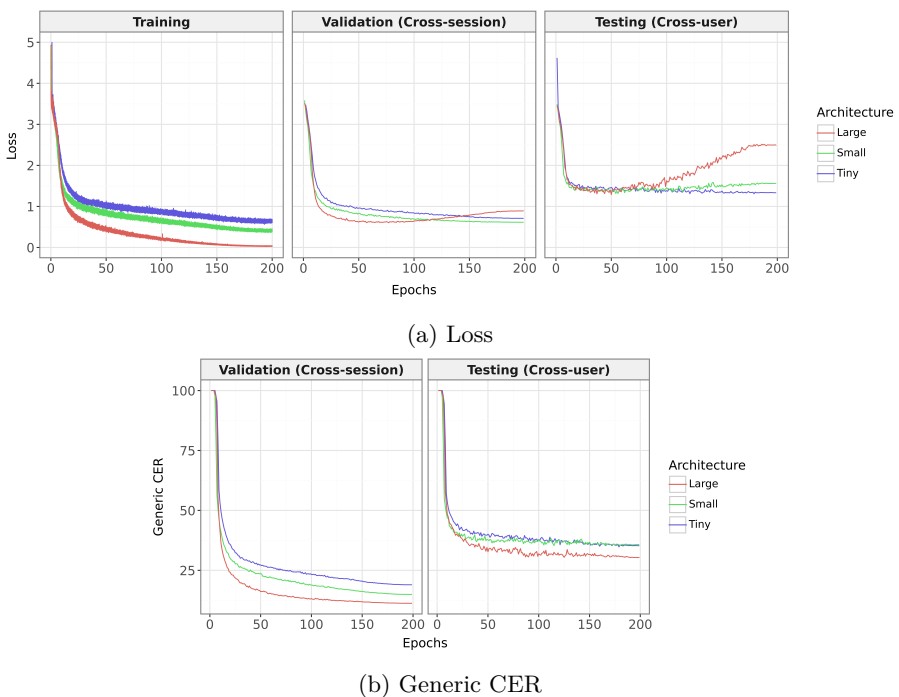

(a) Loss

(b) Generic CER

Figure 4: Supervised training sample training curves for the TINY, SMALL, LARGE architecture. Validation is done with unseen sessions from the training users while testing is done across unseen sessions from the unseen users.

**Accelerated streaming inference** In a more sophisticated implementation of streaming inference, one could further improve the performance of the models by using optimizations like KV Caching (Pope et al., 2023) or designing custom kernels. These techniques are outside the scope of our work.

Table 19: Hyperpameter for baseline experiment in Table 18.

| | | |
|---|---|---|
| Data | Input sEMG channels | 32 |
| | Window length | 8000 |
| | Padding | $[1800, 200]$ |
| Time-CNN | Featurizer channels | $[384, 384, 384]$ |
| | Featurizer kernels | $[11, 5, 5]$ |
| | Featurizer strides | $[5, 2, 2]$ |
| | CNN Encoder input channels | 384 |
| | CNN Encoder channels | $[512, 512, 512, 256]$ |
| | CNN Encoder kernels | $[5, 5, 5, 5]$ |
| | CNN Encoder strides | $[1, 1, 1, 1]$ |
| | Linear Decoder input size | 256 |
| LSTM | Featurizer channels | $[128, 128, 128]$ |
| | Featurizer kernels | $[11, 3, 3]$ |
| | Featurizer strides | $[5, 2, 2]$ |
| | LSTM Encoder input channels | 128 |
| | LSTM Encoder hidden dim | 128 |
| | LSTM Encoder num. layers | 10 |
| | Linear Decoder input size | 256 |
| ViT (d=*) | Featurizer channels | $[128, 64, 64]$ |
| | Featurizer kernels | $[5, 2, 2]$ |
| | Featurizer strides | $[5, 2, 2]$ |
| | Encoder feed-forward ratio | 4 |
| | ViT Encoder hidden size | depends (see name) |
| | Num. Encoder layers | 8 |
| | ViT Encoder attentions heads | 16 |
| | ViT Encoder hidden, attention and activation dropout | 0.2 |
| | ViT Encoder feature projection dropout | 0.2 |
| | ViT Encoder final layer dropout | 0.2 |
| | ViT Encoder causal attention | True |
| | Linear Decoder input size | 256 |
| Task | Tokenizer | Character-level |
| | Vocab size | 99 |
| Training | Epochs | 200.0 |
| | Effective batch size | 640 |
| Optimizer | Learning rate schedule | linear warmup + cosine decay |
| | Learning rate warmup ratio | 0.05 |
| | CTC zero infinity | True |
| | Gradient clipping | 0.1 |
| Software | Torch version | 2.3.1+cu121 |
| | Transformers version | 4.36.2 |

Table 20: Varying hyperpameter for the baseline experiment in Table 18.

| Architecture | Optimizer weight decay | Optimizer learning rate |
|---|---|---|
| Time-CNN | 0.0 | 0.003 |
| LSTM | 0.2 | 0.0003 |
| ViT (d=128) | 0.2 | 0.001 |
| ViT (d=256) | 0.2 | 0.001 |
| ViT (d=512) | 0.2 | 0.0003 |

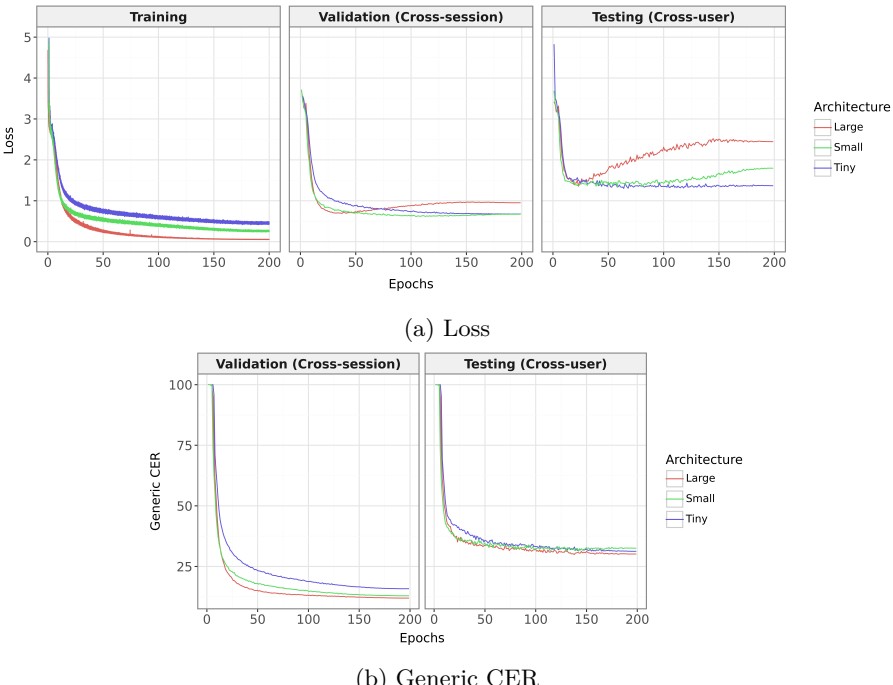

(a) Loss

(b) Generic CER

Figure 5: Distillation training sample training curves for the TINY, SMALL, LARGE architecture. Validation is done with unseen sessions from the training users while testing is done across unseen sessions from the unseen users.

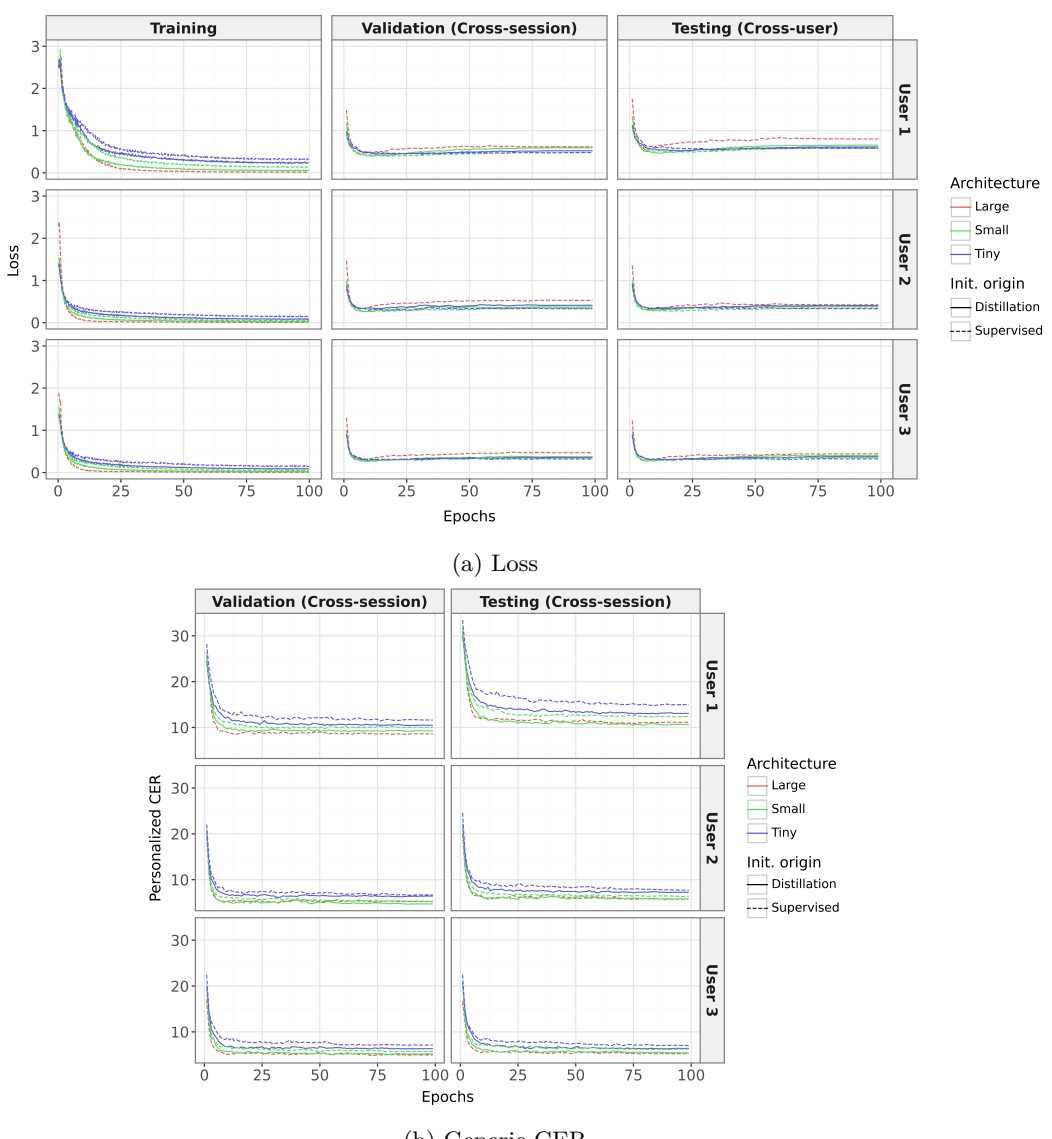

(a) Loss

(b) Generic CER

Figure 6: Personalization training sample training curves for the Tiny, Small, Large architecture. For the Tiny and Small architectures we show the separate curves for initializing the model coming from supervised training and distillation training. We show the training curves for three of the eight personalization users. Both validation and testing is done with unseen sessions from the (single) training user.

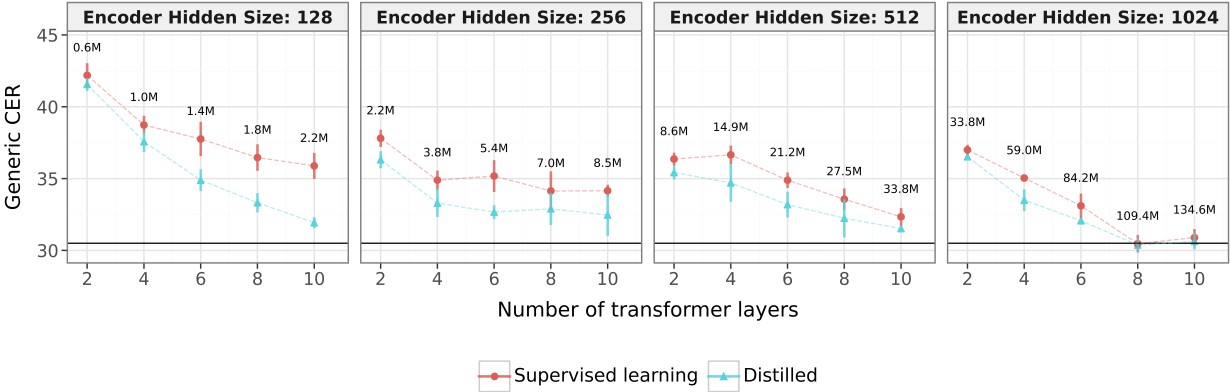

Figure 7: Supervised learning and distilled results on the *emg2qwerty* dataset across multiple model hidden representation size and number of layers. The total number of parameters of the networks are annotated for each configuration. The performance of the teacher model (LARGE architecture) for the distillation training is highlighted by the horizontal line. The vertical bars denote standard deviation across 6 seeds.

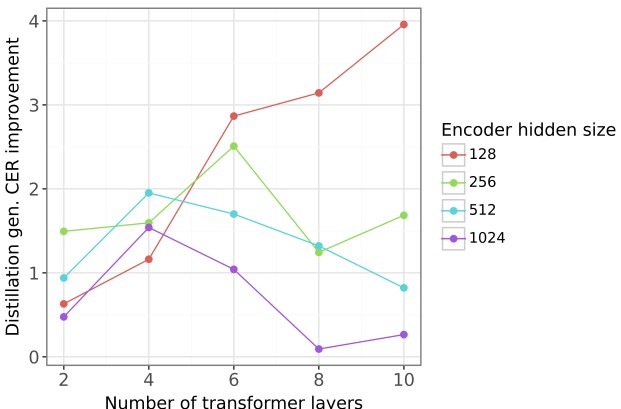

Figure 8: Performance improvement from distillation over supervised learning on the *emg2qwerty* dataset across multiple model hidden representation size and number of layers.

Table 21: Numerical generic CER and their standard deviation for supervised learning analysis. Hyperparameters provided in Section B.2. The specific named architectures from the main text are highlighted in bold.

| Architecture | Encoder hidden size | Encoder layers | Model params | Generic CER |
|---|---|---|---|---|
| | 128 | 2 | 631523 | 42.19 ±0.85 |
| | 128 | 4 | 1028067 | 38.73 ±0.64 |
| | 128 | 6 | 1424611 | 37.76 ±1.20 |
| | 128 | 8 | 1821155 | 36.46 ±0.92 |
| **Tiny** | **128** | **10** | **2217699** | **35.88 ±0.91** |
| | 256 | 2 | 2229091 | 37.81 ±0.60 |
| | 256 | 4 | 3808611 | 34.89 ±0.67 |
| **Small** | **256** | **6** | **5388131** | **35.18 ±1.11** |
| | 256 | 8 | 6967651 | 34.14 ±1.37 |
| | 256 | 10 | 8547171 | 34.15 ±0.42 |
| | 512 | 2 | 8569955 | 36.36 ±0.44 |
| | 512 | 4 | 14874723 | 36.65 ±0.65 |
| | 512 | 6 | 21179491 | 34.89 ±0.54 |
| | 512 | 8 | 27484259 | 33.56 ±0.75 |
| | 512 | 10 | 33789027 | 32.34 ±0.62 |
| | 1024 | 2 | 33834595 | 37.00 ±0.36 |
| | 1024 | 4 | 59027043 | 35.04 ±0.19 |
| | 1024 | 6 | 84219491 | 33.10 ±0.85 |
| **Large** | **1024** | **8** | **109411939** | **30.47 ±0.61** |
| | 1024 | 10 | 134604387 | 30.89 ±0.60 |

Table 22: Numerical generic CER and their standard deviation for distillation analysis. Encoder hidden sizes and layers represents the student encoder sizes. The teacher follows the LARGE architecture. Hyperparameters are provided in Section B.3

| Student arch. | Student encoder hidden size | Student encoder layers | Model params | Generic CER |
|---|---|---|---|---|
| | 128 | 2 | 631523 | 41.56 ±0.45 |
| | 128 | 4 | 1028067 | 37.57 ±0.72 |
| | 128 | 6 | 1424611 | 34.89 ±0.76 |
| | 128 | 8 | 1821155 | 33.32 ±0.67 |
| **Tiny** | **128** | **10** | **2217699** | **31.93 ±0.39** |
| | 256 | 2 | 2229091 | 36.31 ±0.60 |
| | 256 | 4 | 3808611 | 33.29 ±0.98 |
| **Small** | **256** | **6** | **5388131** | **32.67 ±0.48** |
| | 256 | 8 | 6967651 | 32.89 ±1.11 |
| | 256 | 10 | 8547171 | 32.47 ±1.45 |
| | 512 | 2 | 8569955 | 35.42 ±0.48 |
| | 512 | 4 | 14874723 | 34.70 ±1.32 |
| | 512 | 6 | 21179491 | 33.19 ±0.91 |
| | 512 | 8 | 27484259 | 32.24 ±1.34 |
| | 512 | 10 | 33789027 | 31.52 ±0.23 |
| | 1024 | 2 | 33834595 | 36.52 ±0.19 |
| | 1024 | 4 | 59027043 | 33.50 ±0.75 |
| | 1024 | 6 | 84219491 | 32.06 ±0.22 |
| **Large** | **1024** | **8** | **109411939** | **30.38 ±0.46** |
| | 1024 | 10 | 134604387 | 30.63 ±0.56 |

Table 23: Numerical generic CER and their standard deviation for the personalization results with varying model initialization. The initialization origin column represents whether the generic model used as initialization for the personalized model has been trained through *Supervised* learning (i.e., without distillation loss) or through *Distillation*. The initialization seed column represents distinction between different models with the same architecture, but trained on different seeds or different hyperparameters, selected by choosing the best validation performance among the all the models trained. Hyperparameters provided in Section B.4.

| Architecture | Init. origin | Init. seed | Model params | Generic CER |
|---|---|---|---|---|
| Tiny | Distillation | seed A | 2217699 | 8.64 ±0.04 |
| Tiny | Distillation | seed B | 2217699 | 8.65 ±0.02 |
| Tiny | Distillation | seed C | 2217699 | 8.91 ±0.06 |
| Tiny | Supervised | seed A | 2217699 | 9.72 ±0.13 |
| Tiny | Supervised | seed B | 2217699 | 9.91 ±0.13 |
| Tiny | Supervised | seed C | 2217699 | 10.36 ±0.02 |
| Small | Distillation | seed A | 5388131 | 7.07 ±0.06 |
| Small | Distillation | seed B | 5388131 | 7.02 ±0.03 |
| Small | Distillation | seed C | 5388131 | 7.16 ±0.01 |
| Small | Supervised | seed A | 5388131 | 7.94 ±0.06 |
| Small | Supervised | seed B | 5388131 | 9.78 ±0.07 |
| Small | Supervised | seed C | 5388131 | 9.20 ±0.06 |
| Large | Supervised | seed A | 109411939 | 6.81 ±0.07 |
| Large | Supervised | seed B | 109411939 | 6.48 ±0.02 |
| Large | Supervised | seed C | 109411939 | 6.54 ±0.08 |

Table 24: Inference speed of highlighted model sizes (per 4 second window).

| Architecture | Encoder hidden size | Encoder layers | Params | Inference speed (ms) |
|---|---|---|---|---|
| TINY TRANSFORMER | 128 | 10 | 2.2M | 6.1 |
| SMALL TRANSFORMER | 256 | 6 | 5.4M | 5.7 |
| LARGE TRANSFORMER | 1024 | 8 | 109M | 27.0 |

