# OpenReview forum: "Scaling and Distilling Transformer Models for sEMG"
_TMLR — Accepted by TMLR_

### Review · Reviewer_nGrz · 2025-04-05

**Summary Of Contributions:**

This paper investigates the use of small transformer models (2.2M params) on small-scale datasets such as sEMG, which previously failed on sEMG due to the lack of data quantity and diversity, which limited sEMG to old machine learning or deep learning models. They show that with a small amount of data (100 users), transformer models can learn effective representations from the raw sEMG data directly (without the need for hand-crafted feature extraction) and perform well on tasks. They also show, that scaling the transformer model size achieves better results (+20%), a finding consistent with the current literature of transformer models. Based on this finding, they scale the transformer in size, train on sEMG, and then distill a smaller version with 50x less parameters (with only a small drop in performance), which makes it fast and ready to be deployed on edge devices. Finally, they evaluate on more realistic settings from held-out unseen users (zero-shot setting) rather than held-out seen trials from users (where the trial setting is different), and from the (original) noisy, subject-variant sEMG signals. They also promise to release code such that future works can build on it.

**Audience:**

Yes

**Broader Impact Concerns:**

No issues

**Claims And Evidence:**

No

**Requested Changes:**

I would request the authors to address W2 and W3, as they are the most important to me. These are core experiments that should be conducted.

**Strengths And Weaknesses:**

Strengths:
- The paper shows that Transformer models are effective for sEMG, and show that better performance can be achieved by scaling up the transformer, even though the data scale remains small.
- The authors distill the transformer into a smaller size, making it applicable for on-edge devices which is important for such settings,
- Evaluation is performed on a more realistic setting, which I appreciate.

Weaknesses:

- [W1] One of the contributions is unclear to me. Transformers were successful on small-scale sEMG datasets. The authors mention in the related work: "Some recent work has begun to explore the application of transformer-based models [cited works] and vision transformers (ViTs) to sEMG data [cited works] but these efforts are also limited to small-scale datasets and simpler tasks". This means that the contribution is probably investigating transformer models on larger scale datasets (which should perform better since the data scale is larger)? I am confused about this contribution where the authors show that transformer models can be effective for sEMG tasks even
when considering small-scale datasets (compared to standard deep learning datasets).
- [W2] While the authors show that Transformer models are effective for sEMG data, It is unclear to me *why* do they work. Is it because of the hybrid transformer architecture (the 3 CNN layers the authors use at the beginning), or the alleviation of hand-crafted features (which I assume is the case), or the data augmentation...etc. I would appreciate ablation studies on this.
- [W3] The authors mention that prior works (Table 1) fail. However, this is not shown quantitatively for the same dataset and experimental setting that the authors consider. There are no comparisons with respect to other works show in Table 1, in the results in Tables 2 and 3. Instead, the authors show results for their experiments only. I understand that replicating results of other works might be difficult, especially if the code is not provided. Therefore, I recommend the authors to perform their own replication of some prior works, as additional baselines, with exactly the same dataset, experimental and evaluation setting. I would like that the authors focus on Transformer-based and hybrid-based models (for example, Rahimian et al., 2021, Godoy et al., 2022, Chen et al., 2023, Zabihi et al., 2023, Zhang et al., 2023, (Liu et al., 2024, Putro et al., 2024) as this is the focus of the proposed method.

---

> ### Author Response · Authors · 2025-04-11
> **Authors’ response to Reviewer nGrz**
>
> Thank you for dedicating your time and effort to reviewing our manuscript. We greatly appreciate your recognition of the value of our work, specifically that better performance can be achieved by scaling up model scale and that this performance can be effectively transferred to smaller scale models through distillation for on-edge device inference. We are glad that you appreciate our effort towards evaluating our approaches on realistic settings with cross-user evaluation.
>
>
> *Note:* Please note that for the requested changes addressed below were added to the manuscript, however we refrain from revising the OpenReview manuscript out of respect for the other reviewers, following the [TMLR guidelines](https://jmlr.org/tmlr/editorial-policies.html).
> > Authors can respond to a review as soon as it is posted, however we recommend waiting until all 3 reviews have been submitted before submitting any revised version of the PDF manuscript.
>
>
> **[W1]** We apologize for the oversight  – the claim regarding small transformers having not been previously applied to sEMG is incorrect, and left over from an earlier draft before a deeper literature review uncovered the prior work we discuss later in the manuscript. We reformulated our statement in the Introduction to focus on the model scaling and data-driven part of our contribution, as follows:
> > We show that a simple convolution and transformer architecture applied directly to the sEMG signal is a very effective approach for sEMG decoding, outperforming previous SOTA by 20% (absolute) on the emg2qwerty task. We further show that the performance of the transformer models improves with model scale,  enabling us to improve over the SOTA performance by an additional 5% (absolute) by increasing the number of parameters from 2.2M to 109M. This differs from other works which mainly focus on small model scales (<1M parameters  [Zabihi et al., 2023; Rahimian et al., 2021]) applied to preprocessed sEMG features (e.g. spectrograms [Sivakumar et al., 2024; CTRL-labs at Reality Labs, 2024], and other handcrafted time and frequency features [Putro et al., 2024]).
>
> **[W2]** We appreciate the reviewer’s suggestion. We have performed the ablation experiment on encoder architecture (TDS, Transformer), featurizer architecture (Raw sEMG Signal+CNN, Spectrogram+MLP) and data augmentation techniques (channel rotation). We show in Table 1 that changing the featurizer from a preprocessed type such as Spectrogram+MLP to a data-driven Raw sEMG Signal+CNN yields an 10.94 CER improvement for the TDS encoder on the emg2qwerty task, and changing the TDS encoder for a Transformer encoder yields a further 8.88 CER improvement.
>
> |                             |                     |              |                               |
> |------------------------|------------------|-------------------------|---------------------------|
> |  **Featurizer type** |                     | Raw EMG Signal+CNN          | Spectrogram+MLP |
> | **Encoder type** | TDS             | 45.84±0.06            | 56.78±0.17             |
> |                             | Transformer | 36.96±0.95            | 45.15±1.17            |
>
>
> **Table 1**: Encoder type and featurizer type ablation
>
> Furthermore, we show in Table 2 the contribution of the channel rotation data augmentation and see that it leads to a 0.96 CER improvement on the Signal+CNN featurizer with Transformer encoder.
>
> |                      |  |                     |
> |------------------|-----------------------------|-----------------|
> | **Channel rotation** | None                         | ±1               |
> |    | 37.88±0.82                | 36.92±0.82  |
>
>
> Table 2: Channel rotation ablation for Signal+CNN featurizer with Transformer encoder
>
> So to answer the reviewer’s question about why the architecture works for the problem, these results show the independent value of our modeling choices. We will add these tables and discussion to the manuscript.
>
> **[W3]** We apologize for the confusion. Our intent is to highlight that prior work fails to evaluate models on the more ecologically realistic cross-user setup we use, but we do not intend to make a claim on whether those models would succeed in that setting. We will revise the manuscript in section 2.1 to clarify this:
> > However, many prior works neglect this crucial consideration, instead opting to test their models solely on held-out trials from the same individuals contained within the training set, thus failing to provide a meaningful assessment of the model's ability to generalize across users.

---

### Review · Reviewer_gqjk · 2025-04-16

**Summary Of Contributions:**

This paper presents experiments and results on a) training transformer models on sEMG (Surface Electromyographic) datasets, b) performing model distillation on such data and models. Experiments are performed on the emg2qwerty dataset, with around 346 hours of data of more than 100 users. The non-transfomer baseline is taken from the original emg2qwerty paper (Sivakumar et al); settings for the transformer include CTC loss, the AdamW optimizer, and a grid search on architectural hyperparameters (section 3.2). Results are reported in subject-independent settings (i.e. the set of subjects whose data are used for training/testing are disjoint).

**Audience:**

Yes

**Broader Impact Concerns:**

Given the major weakness described above, the impact of this paper would be quite limited.

**Claims And Evidence:**

No

**Requested Changes:**

Since this paper provides a substantial improvement over the baseline, I think it could be interesting, but it needs to be rewritten regarding the premise (critical request for acceptance). The mere fact that a transformer works on this kind of data is not really interesting.

**Strengths And Weaknesses:**

Strengths:
- Results in subject-independent settings are interesting and relevant, since sEMG data differs quite a lot between subjects.
- Improvement over the baseline (table 2) is substantial.
- Decent writing, no major technical concerns.

Weaknesses:
- In my opinion the major weakness is the flawed initial claim that training transformer-based models on sEMG data is new. A quick internet search produces, for example:
* Montazerin et al, Transformer‑based hand gesture recognition from instantaneous to fused neural decomposition of high‑density EMG signals
* Scheck et al, Multi-Speaker Speech Synthesis from Electromyographic Signals by Soft Speech Unit Prediction

all using some kind of transformer model and much less data than the present study. Both works are not cited.
- Also, there is no real technical contribution: the techniques (transformer, distillation) are well-known, just the data is new.

---

> ### Author Response · Authors · 2025-04-17
> **Authors’ response to Reviewer gqjk**
>
> Thank you for dedicating your time and effort to reviewing our manuscript. We are glad that you appreciate our effort towards evaluating our approaches on realistic settings with cross-user evaluation and find that our results offer substantial improvement over the baseline.
>
> **Note**:  Please note that for the requested changes addressed below were added to the manuscript, however we refrain from revising the OpenReview manuscript out of respect for the other reviewers, following the [TMLR guidelines](https://jmlr.org/tmlr/editorial-policies.html).
> > Authors can respond to a review as soon as it is posted, however we recommend waiting until all 3 reviews have been submitted before submitting any revised version of the PDF manuscript.
>
> **Flawed initial claim that training transformer-based models on sEMG data is new**
>
> We apologize for the oversight  – the claim regarding transformers having not been previously applied to sEMG is incorrect, and left over from an earlier draft before a deeper literature review uncovered the prior work we discuss later in the manuscript. We have added the proposed [1,2] to our references of transformer-based architectures in Section 2.1. We have also reformulated our statement in the Introduction to focus on the model scaling and data-driven part of our contribution, as follows:
> > We show that a simple convolution and transformer architecture applied directly to the sEMG signal is a very effective approach for sEMG decoding, outperforming previous SOTA by 20% (absolute) on the emg2qwerty task. We further show that the performance of the transformer models improves with model scale, enabling us to improve over the SOTA performance by an additional 5% (absolute) by increasing the number of parameters from 2.2M to 109M. This differs from other works which mainly focus on small model scales (<1M parameters [1]) applied to preprocessed sEMG features (e.g. spectrograms, and other handcrafted time and frequency features).
>
> This reformulated contribution statement highlights that by taking a simple modeling approach, and scaling up the capacity of the model yields considerable performance improvements (5% improvement by scaling 2.2M ->109M parameters), and that some of those performance improvements can be retained in edge device settings by distilling into a smaller model capacity (50x parameter reduction with only 1.5% degradation).
>
> **Transformers are well-known, just the data is new**
> As mentioned in the previous point, we’ve reformulated the contributions to clarify that our contribution is not just the application of the transformer architecture on the sEMG data, but rather demonstrating that scaling transformer architectures improves performance on the sEMG tasks for unseen users even though these datasets are relatively small compared to the scale of datasets using in many other deep-learning settings. We also show that these larger transformer models can be extensively distilled (50x fewer parameters) with a marginal performance regression (1.5% degradation).
>
> **Distillation are well-known, just the data is new**
> Knowledge distillation is something known in many fields of deep learning. However, we have not seen it adequately studied in the sEMG field, where it is particularly valuable for the potential of streaming on-device performance. We will add the following paragraph discussing those related works in the background section.
> > Some past studies began to explore the application of distillation techniques for models trained on sEMG datasets. For instance, [3] distilled a ResNet model ensemble into a single model instance with good success, however their evaluation is limited to intra-user and intra-session generalization across only five users. In contrast, our work: (i) focuses on cross-user evaluation, (ii)  trains on a significantly larger dataset (346 hours compared to 10 hours) and model (~130M parameters compared to ~35M parameters), (iii) distills the model extensively (up to 180x size reduction range compared to up to 10x range) which contributes to a better understanding of different regimes where the distillation works and where it doesn’t. Additionally, some research has investigated cross-modal knowledge distillation in context of sEMG tasks. Notably, [4] examined distillation between sEMG and hand gestures, while [5] explored distillation between sEMG and ultrasound. In contrast, our work concentrates on distilling knowledge from models trained exclusively on sEMG signals.
>
> We believe that the above additions highlight that our evaluation of distillation on sEMG signal is a useful and non-trivial contribution to the field.
>
> **References** See references [1,2,3,4,5] in the follow up comment

---

> > ### Author Response · Authors · 2025-04-17
> > **References for "Authors’ response to Reviewer gqjk"**
> >
> > **References**
> >
> > [1] Montazerin, Mansooreh, et al. "Transformer-based hand gesture recognition from instantaneous to fused neural decomposition of high-density EMG signals." Scientific reports 13.1 (2023): 11000.
> >
> > [2] Scheck, Kevin, and Tanja Schultz. "Multi-speaker speech synthesis from electromyographic signals by soft speech unit
> > prediction." ICASSP 2023-2023 IEEE International Conference on Acoustics, Speech and Signal Processing (ICASSP). IEEE, 2023.
> >
> > [3] Lai, Wenqiang, et al. "Knowledge Distilled Ensemble Model for sEMG-based Silent Speech Interface." IEEE EUROCON 2023-20th International Conference on Smart Technologies. IEEE, 2023.
> >
> > [4] Dai, Qingfeng, et al. "Improved network and training scheme for cross-trial surface electromyography (semg)-based gesture recognition." Bioengineering 10.9 (2023): 1101.
> >
> > [5] Zeng, Jia, et al. "Cross modality knowledge distillation between A-mode ultrasound and surface electromyography." IEEE Transactions on Instrumentation and Measurement 71 (2022): 1-9.

---

### Review · Reviewer_QySn · 2025-05-22

**Summary Of Contributions:**

The paper proposes a transformer network for training on sEMG dataset. Benchmarking on emg2qwerty results in favorable comparisons against baseline methods. The paper shows how distillation can further reduce the parameter count of the network.

**Audience:**

No

**Claims And Evidence:**

No

**Requested Changes:**

- Comparison against the real state-of-the-art methods
- Clear positioning in the literature
- Empirical or theoretical justification for why the proposed method is of interest and be used by practitioners
- Justifications for the claimed contributions- distillation, validation via held-out set have all been done before and cannot be claimed as new contributions.
- Any new insights that the model uncovers regarding the data

**Strengths And Weaknesses:**

While the results presented in the paper seem correct, the claims are not fully substantiated by it. There is plenty of work on transformers, compact transformers and even distilled transformers on sEMG, none of which appear in the baselines that the authors compare against [1,2,3,4 to name a few but there are plenty]. The only addition that I see in the proposed work is benchmarking on the emg2qwerty dataset but because the dataset was released in December last year (Neurips), it is understandable that there wont be any methods benchmarking on it. There is no innovation on learning, dataset, evaluation, etc. and while novelty isn't a criteria for TMLR, I dont see why the paper would be of interest to the readers when there are already so many works in the same space.

[1] Wanga, Zefeng, et al. "Lightweight Transformer for sEMG Gesture Recognition with Feature Distilled Variational Information Bottleneck." (2024).
[2] Zhang, Jiaxuan, et al. "Feasibility analysis of semg recognition via channel-wise transformer." 2022 IEEE 11th Global Conference on Consumer Electronics (GCCE). IEEE, 2022.
[3] Yang, Jehan, et al. "EMGBench: Benchmarking Out-of-Distribution Generalization and Adaptation for Electromyography." Advances in Neural Information Processing Systems 37 (2024): 50313-50342.
[4] Zhang, Jiaxuan, et al. "Movement recognition via channel-activation-wise sEMG attention." Methods 218 (2023): 39-47.

---

> ### Author Response · Authors · 2025-05-30
> **Authors’ response to Reviewer QySn**
>
> Thank you for dedicating your time and effort to reviewing our manuscript. We appreciate your recognition that our results offer substantial improvement over the baseline while not increasing the parameter count.
>
> **[Preface]**
>
> We start this response by responding to the reviewer's comment: “I dont see why the paper would be of interest to the readers when there are already so many works in the same space.”
> **We emphasize that we think such “simple work, done well, and reported clearly” contributions are in fact an ideal fit for TMLR.**
>
> **[RC1] Comparison against the real state-of-the-art methods**
>
> We acknowledge that we are lacking in baselines to connect this work with the rest of the literature. We are in the process of running the experiments and will report back with the results in a week. We respond to this review without these results to engage in discussion for the other requested changes.
>
> **[RC2] Clear positioning in the literature**
>
> We agree that the original draft over-emphasized the uniqueness of compact transformers EMG, and adjusted the wording throughout to pull back on this claim, having received similar feedback from the other reviewers.
>
> As to the positioning in the literature, we believe our work brings a unique emphasis on (a) scaling up the simplest thing, (b) extensive model size reduction investigation (c) and cross-user evaluation that is missing from other work.
>
> **Scaling up the simplest thing** We focus our analysis on leveraging model size of _well-established architectures_ to improve performance. This contrasts with most existing work which focus on small scale (<1M parameters) and complex bespoke domain-specific modifications [1,2,4,12&more]. We’re not aware of other works showing a comprehensive investigation of scaling model size of any architecture and its contribution to performance in the sEMG field. The closest analysis that compares is analysis in [11] and [12], but both investigate fewer than five model sizes for a given setting, all at a small scale (<100k parameters for [11] and <1M parameters for [12]).
>
> **Model size reduction for sEMG** Because scaling up model size might not be feasible for on-device constraints, we investigate the efficacy of knowledge distillation of up to two orders of magnitude size reduction.. This is in contrast to work which uses sophisticated variational objectives [1] or ensemble distillation [8] and only considers size reductions up to 10x (vs 180x in ours).
>
> **Cross-user evaluation** As the work you’ve shared [3] emphasises, cross-user generalization is a critical part of building robust HCI, because the generalization is much harder as shown in [6,9] where both report >30% absolute classification regression between intra-user and inter-user. While cross-user generalization has been studied in the context of sEMG, it remains uncommon, and we have not seen other works that have done distillation on a user-generalization task.
>
> Our positioning in the literature is the gap along those three axes. We have added Table 6 in the manuscript detailing the position of closely related works along these three axes.

---

> ### Author Response · Authors · 2025-05-30
> **Authors’ response to Reviewer QySn (Part 2)**
>
> **[RC3] Empirical or theoretical justification for why the proposed method is of interest and be used by practitioners**
>
> Echoing points from RC2, the work’s strength is in part that it provides a pragmatic recipe for practitioners. First, vanilla transformers can be scaled up for better cross-user performance up to bigger than previously reported in other works. Second, the simplest form of distillation works and can be used to bring scaled models to fit into specific compute constraints with minimal performance degradation (up to 50x). Thirdly, we provide an empirical limit where this approach seems to yield diminishing returns (>50x size reduction).
>
> The complex and bespoke domain-specific modifications in prior work hinder their applicability for real-world practitioners because of implementation difficulties and divergence from standard architectures which are widely used and battle-tested by thousands of researchers across domains and settings.. This is especially true for in-the-wild practitioners, often  neuroscientists who might not be as familiar with deep learning systems. A perfect example of this is the distillation work you refer us to [1], which a) modifies transformer attention with depthwise-followed-by-pointwise convolutional feature projections; b) modifies the MLP with a variational information bottleneck layer; c) introduces changes to the skip connection structure compared to vanilla transformers and d) uses a dual objective distillation setup.
>
> **[RC4] Justifications for the claimed contributions- distillation, validation via held-out set have all been done before and cannot be claimed as new contributions.**
>
> As noted in [RC2], while both settings involve held-out validation, cross-user generalization is a different and more difficult problem relative to intra-user evaluation. Reports of this in the literature are plenty: notably [6,9] both show >30% absolute classification regression between intra-user and inter-user, [10] shows a 40% CER regression, [3] shows significant regressions in Table 1 and 2. We agree that cross-user generalization has been previously in the sEMG field (e.g. [3,6,7]), and we think this should be the default setting due to its greater applicability to real-world settings. However, it is still uncommon and most recent works are to this day evaluating on the Ninapro dataset (e.g. works that you have shared [1, 2, 4] still evaluate within-users). Thus, we do think it is important to highlight that we operate in the more challenging and realistic setting, and we have not seen other works that have done distillation on a user-generalization task.
>
> For a clearer picture, see Table 6 of the updated manuscript to see which related work (a) does cross-user generalization and (b) are using Ninapro databases.
>
> **[RC5] Any new insights that the model uncovers regarding the data**
>
> We agree that new insights may be interesting given this recently released dataset, but this is not in scope for our work, which is focused on generalizable decoding performance rather than learning something new about the sEMG signal.

---

> > ### Author Response · Authors · 2025-05-30
> > **References for "Authors’ response to Reviewer QySn"**
> >
> > **References**
> >
> > [1] Wanga, Zefeng, et al. "Lightweight Transformer for sEMG Gesture Recognition with Feature Distilled Variational Information Bottleneck." (2024).
> >
> > [2] Zhang, Jiaxuan, et al. "Feasibility analysis of semg recognition via channel-wise transformer." 2022 IEEE 11th Global Conference on Consumer Electronics (GCCE). IEEE, 2022.
> >
> > [3] Yang, Jehan, et al. "EMGBench: Benchmarking Out-of-Distribution Generalization and Adaptation for Electromyography." Advances in Neural Information Processing Systems 37 (2024): 50313-50342.
> >
> > [4] Zhang, Jiaxuan, et al. "Movement recognition via channel-activation-wise sEMG attention." Methods 218 (2023): 39-47.
> >
> > [5] Schneider, Steffen, et al. "wav2vec: Unsupervised pre-training for speech recognition." arXiv preprint arXiv:1904.05862 (2019).
> >
> > [6] Ctrl-labs at Reality Labs, et al. "A generic noninvasive neuromotor interface for human-computer interaction." Biorxiv (2024): 2024-02.
> >
> > [7] Scheck, Kevin, and Tanja Schultz. "Multi-speaker speech synthesis from electromyographic signals by soft speech unit prediction." ICASSP 2023-2023 IEEE International Conference on Acoustics, Speech and Signal Processing (ICASSP). IEEE, 2023.
> >
> > [8] Lai, Wenqiang, et al. "Knowledge Distilled Ensemble Model for sEMG-based Silent Speech Interface." IEEE EUROCON 2023-20th International Conference on Smart Technologies. IEEE, 2023.
> >
> > [9] Saponas, T. Scott, et al. "Demonstrating the feasibility of using forearm electromyography for muscle-computer interfaces." Proceedings of the SIGCHI conference on human factors in computing systems. 2008.
> >
> > [10] Sivakumar, Viswanath, et al. "emg2qwerty: A large dataset with baselines for touch typing using surface electromyography." Advances in Neural Information Processing Systems 37 (2024): 91373-91389.
> >
> > [11] Rahimian, Elahe, et al. "Temgnet: Deep transformer-based decoding of upperlimb semg for hand gestures recognition." arXiv preprint arXiv:2109.12379 (2021).
> >
> > [12] Zabihi, Soheil, et al. "Trahgr: Transformer for hand gesture recognition via electromyography." IEEE Transactions on Neural Systems and Rehabilitation Engineering 31 (2023): 4211-4224.

---

> ### Author Response · Authors · 2025-06-06
> **Following up on requested experiments of Rev. QySn**
>
> As promised, we have conducted some baseline experiments to better align our work with existing literature. Following are the results.
>
> We investigated three models: (a) a Time Convolutional Neural Network (Time-CNN), (b) a Long Short-Term Memory (LSTM) network, and (c) a Visual Transformer (ViT).
> * **Time-CNN**: This model uses a raw sEMG+CNN featurizer (same as our Transformer architecture) and time-wise convolutional layers as encoder, each with a kernel size of 5 and a stride of 1 with channel dimensions of [512,512,512,256].
> * **LSTM**: This model uses a raw sEMG+CNN featurizer (same as our Transformer architecture) and a uni-directional LSTM network as encoder with 10 layers and dimension of 256.
> * **ViT**: This model employs the same transformer encoder architecture (same as our Transformer architecture). However, the featurization is CNN-based applied on raw sEMG signal, but it uses non-overlapping 10ms patches that include all EMG channels. We use a 8 layer encoder, and we do a simple sweep across the dimension of the transformer to peek into the scalability of the model.
> The results of these experiments are presented in Table 1.
>
> **Table 1: Cross-user performance of simple architectural baselines on the emg2qwerty dataset.**
> |**Architecture**   | **Parameters** | **CER (%)**    |
> |--------------------|----------------|----------------|
> | TDS-ConvNet        | 5.3M           | 55.57          |
> | Time-CNN         | 5.9M           | 70.14 ±0.35    |
> | LSTM               | 5.3M           | 45.96 ±1.02    |
> | ViT (d=128)        | 1.8M           | 37.29 ±1.57    |
> | ViT (d=256)        | 6.9M           | 35.29 ±0.66    |
> | ViT (d=512)        | 27M            | 34.73 ±0.99    |
> | Tiny Transformer       | 2.2M           | 35.9 ±0.9      |
> | Small Transformer      | 5.4M           | 35.2 ±1.1      |
> | Large Transformer      | 109M           | 30.5 ±0.6      |
>
> In Table 1, we see that the transformer model demonstrates similar performance to Vision Transformer (ViT) architecture for similar model sizes (e.g., 6.9M and 5.4M both are about 35 CER). In contrast, the Time-CNN model exhibits significantly poorer performance compared to the other architectures. We attribute this to the Time-CNN's finite (and small) receptive field, which limits its ability to utilize past context from the sequence for predictions. Although TDS-ConvNet is also based on CNN layers, it is less affected by this limitation due to its design, which is tailored to maintain larger receptive fields [1] across the sequence.
>
> The LSTM (with Raw sEMG+CNN featurizer) model performs reasonably well, surpassing the previous TDS-ConvNet baseline (with Spectrogram+MLP featurizer). However, this improvement may largely be due to the enhanced featurizer (Raw sEMG+CNN vs Spectrogram+MLP). From the ablation experiments in Table 4 of the Section 5 in the revised manuscript, we see that a Raw+CNN featurizer with TDS encoder performs about the same as the Raw sEMG+CNN featurizer with LSTM encoder model investigated here (i.e., about 45 CER).
>
> We have added Appendix C.1. in the revised manuscript to include these new results and add a mention of them in Section 4.1. Look for edits in red.
>
> **References**
>
> [1] Hannun, Awni, et al. "Sequence-to-sequence speech recognition with time-depth separable convolutions." arXiv preprint arXiv:1904.02619 (2019).

---

### Author Response · Authors · 2025-05-30
**General response to reviewers and AE**

We would like to express our gratitude to all the reviewers for their diligent efforts in reviewing our manuscript. We sincerely appreciate the valuable feedback provided in the initial reviews, we’ve updated several parts of the manuscript in light of those and the manuscript is all the better for it. In this response to all, we will provide an overview of the updates made to the manuscript, leaving reviewer specific discussion in their specific threads.

**We invite the reviewers to engage in discussion and give us feedback on whether the work we’ve done to address their concerns and requested changes are sufficient.**

## Broad scope adjustments

**Attention-based models** As reviewer nGRz, gqjk and QySn have pointed out, the initial scope of our contribution regarding the use of attention based models (vanilla and vision transformers) was not well calibrated regarding some related works which we had missed during our initial literature review. We have adjusted the background section and contributions in the manuscript to focus on the our methodology of scaling up the simplest thing, which we have identified as missing from the existing literature:

> We start by applying a simple convolution and transformer architecture on the emg2qwerty task and outperform previous CNN-based SOTA by 20% (absolute). Then, we show that the performance of the transformer can be further improved with model scale, enabling us to improve over the SOTA performance by an additional 5% (absolute) by increasing the number of parameters from 2.2M to 109M. This differs from other works which mainly focus on small model scales (<10M) parameters with complex non-standard deep learning methodology.

**Knowledge distillation in sEMG** As reviewer gqjk and QySn have pointed out, knowledge distillation has been applied in the sEMG setting, and this is not something we had fully captured in the description of our contribution and background. However, we still find our work provides useful insights to practitioners in a few regards: (a) focusing on the simplest possible method (logit-distillation) (b) extensive investigation of the size reduction axis and (c) cross-user evaluation.
As with our focus on vanilla transformers, we focus on vanilla logit-based distillation. We show that this simplest approach works, and works well.
Our large size reduction range shows that logit model distillation can be done effectively up to 50x parameter count reduction. We aren’t aware of any work that measured the performance dropoff of distillation on EMG at scales up to 180x smaller.
The cross-user evaluation is also a critical part of the setup, because intra-user generalization is not measuring real-world applicability of those methods. We have not seen other works that have done distillation on a user-generalization task or at our scale.
To better reflect these points, we updated the contribution description around distillation and added a paragraph in the background section.

> We analyze simple logit model distillation across model size reduction ranging from no reduction to 180x parameter reduction. We show that training larger transformer models followed by distillation into smaller models substantially outperforms direct training of the small-sized transformer without distillation across all size reduction investigated, and that we can reduce the parameter count of the transformer model by up to 50x before seeing significant performance degradation (<1.5% absolute).

**Putting it all together, visually**

To better visualize the landscape of related works in the field, we’ve added Table 6 which summarizes the closest related work to ours along our 3 contribution axes, i.e., model scale, distillation reduction scale and cross-user generalization evaluation. It can be found in Appendix A.

## Ablation studies

Following suggestion from reviewer nGrz, we have performed the ablation experiment on encoder architecture (TDS, Transformer), featurizer architecture (Raw sEMG Signal+CNN, Spectrogram+MLP) and data augmentation techniques (channel rotation and input masking). These can all be found in the new Analysis section (Section 5).

## Other changes
* [nGrz, gqjk,QySn] Updated the abstract to better reflect the contributions
* [nGrz, gqjk,QySn] Updated parts of the introduction to better ease into the contribution
* [nGrz] Prior work description clarification (fail vs. does not do)
* [nGrz, gqjk,QySn] Updated title of section 4.1
* [nGrz, gqjk,QySn] Updated the conclusion to better wrap up our contributions

---

### Author Response · Authors · 2025-06-06
**Updated manuscript**

We have updated the manuscript with a [follow up experiment](https://openreview.net/forum?id=hFPWThwUiZ&noteId=rf5ueUSt6e) requested by Reviewer QySn including additional LSTM, convolutional network, and ViT baselines. Our approach matches or outperforms these additional baselines. The changes are highlighted in red.

Please let us know if you have follow up concerns or questions.

---

### Decision · Action_Editor_8KF3 · 2025-06-21

**Recommendation:** Accept with minor revision

**Additional Comments:**

The revised manuscript contains typos (e.g., "these contributions provides") and low-resolution graphics with oversized title fonts. Reviewer QySn also noted that the manuscript still reads like a technical report. I therefore recommend a linguistic and stylistic revision. Additionally, as the reviewer still finds the justification for the proposed method unclear, this aspect should be further clarified.

**Audience:**

Yes

**Audience Explanation:**

Their simple approach outperforms other baselines notably. The paper further provides a nice overview of other existing approaches.

**Claims And Evidence:**

Yes

**Claims Explanation:**

As also stated by Reviewer gqjk, the contribution in essence is that the paper

> provides a simple and pragmatic recipe for practitioners to apply in realworld setting

This is supported by empirical evidence.